# PTQ4ARVG: Post-Training Quantization for AutoRegressive Visual Generation Models

**Xuewen Liu**[1,2], **Zhikai Li**[1,*] **Jing Zhang**[1,2], **Mengjuan Chen**[1], **Jianquan Li**[1], **Qingyi Gu**[1,*]

[1]Institute of Automation, Chinese Academy of Sciences
[2]School of Artificial Intelligence, University of Chinese Academy of Sciences
`{liuxuewen2023, zhikai.li, qingyi.gu}@ia.ac.cn`

## ABSTRACT

AutoRegressive Visual Generation (ARVG) models retain an architecture compatible with language models, while achieving performance comparable to diffusion-based models. Quantization is commonly employed in neural networks to reduce model size and computational latency. However, applying quantization to ARVG remains largely underexplored, and existing quantization methods fail to generalize effectively to ARVG models. In this paper, we explore this issue and identify three key challenges: (1) severe outliers at channel-wise level, (2) highly dynamic activations at token-wise level, and (3) mismatched distribution information at sample-wise level. To these ends, we propose PTQ4ARVG, a training-free post-training quantization (PTQ) framework consisting of: (1) Gain-Projected Scaling (GPS) mitigates the channel-wise outliers, which expands the quantization loss via a Taylor series to quantify the gain of scaling for activation-weight quantization, and derives the optimal scaling factor through differentiation. (2) Static Token-Wise Quantization (STWQ) leverages the inherent properties of ARVG, fixed token length and position-invariant distribution across samples, to address token-wise variance without incurring dynamic calibration overhead. (3) Distribution-Guided Calibration (DGC) selects samples that contribute most to distributional entropy, eliminating the sample-wise distribution mismatch. Extensive experiments show that PTQ4ARVG can effectively quantize the ARVG family models to 8-bit and 6-bit while maintaining competitive performance. Code is available at http://github.com/BienLuky/PTQ4ARVG

## 1 INTRODUCTION

Recently, motivated by the success of autoregressive generation in large language models (LLMs) (Touvron et al., 2023; Liu et al., 2024a) and the rising demand from multimodal tasks (Ramesh et al., 2021; Wang et al., 2021), research in visual generation has shifted back toward autoregressive paradigms. A growing number of autoregressive visual generation (ARVG) models (Tian et al., 2024; Wang et al., 2024; Yu et al., 2024; He et al., 2025; Yao et al., 2024; Chen et al., 2025; Liu et al., 2024b; Li et al., 2024b) have emerged, surpassing state-of-the-art diffusion models (Li et al., 2025) in image generation. However, the large model sizes and iterative token predictions impose substantial memory and computational overhead, significantly limiting their applicability and generalization. For instance, VAR-d30 (Tian et al., 2024), RAR-XXL (Yu et al., 2024), and MAR-Huge (Li et al., 2024b) contain 2B, 1.5B, and 1B parameters, respectively, while the 3B-parameter PAR (Wang et al., 2024) model requires more than **3** seconds to generate a single image.

Quantization discretizes floating-point parameters into integers, thereby reducing both model size and computational cost. It is typically categorized into Quantization-Aware Training (QAT) (Li & Gu, 2023; Liu et al., 2024c; Esser et al., 2019) and Post-Training Quantization (PTQ) (Li et al., 2022; Liu et al., 2024d; 2025; Li et al., 2023b). While QAT maintains performance through full-model retraining, it demands large amounts of training data and expensive resources. In contrast, PTQ requires only a small calibration and does not rely on model training, making it more desirable for compressing and accelerating ARVG models.

---

*Corresponding author: {zhikai.li, qingyi.gu}@ia.ac.cn.

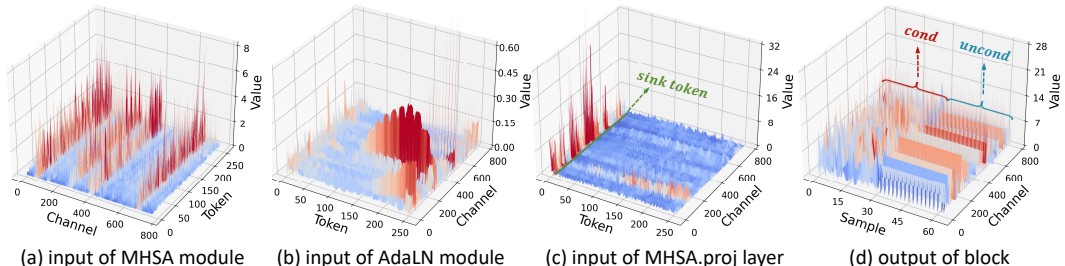

(a) input of MHSA module    (b) input of AdaLN module    (c) input of MHSA.proj layer    (d) output of block

Figure 1: **Challenges of ARVG quantization.** (a) Severe outliers at channel-wise level. (b)(c) Highly dynamic activations at token-wise level. (d) Mismatched distribution information at sample-wise level. Data from the RAR-B block "blocks.23".

To explore effective PTQ methods for ARVG models, we first examine the potential constraints and challenges involved: ① **Severe outliers at channel-wise level.** The activations adjusted by the AdaLN module, i.e., the inputs to the Multi-Head Self-Attention (MHSA) and Feedforward Network (FFN), suffer from significant outliers at the channel-wise level. As shown in Fig. 1(a), the activations exhibit extremely wide ranges and substantial variation across channels. ② **Highly dynamic activations at token-wise level.** To preserve the bidirectional token dependencies (Tian et al., 2024) of ARVG, the input to the AdaLN module contains positional embedding information, which displays highly dynamic distributions along the token dimension, as shown in Fig. 1(b). Additionally, as ARVG employs conditional information as the initial token, which are sensitive to quantization, it results in the presence of *sink* tokens in the activations of all linear layers within MHSA and FFN, as shown in Fig. 1(c). ③ **Mismatched distribution information at sample-wise level.** As shown in Fig. 1(d), the network activations exhibit high similarity across input samples, particularly for unconditional samples. This sample-wise redundancy leads to mismatched calibration of quantization parameters. Notably, these challenges emerge across different layers of the network, as shown in Fig. 2, necessitating layer-specific quantization strategies.

Despite previous efforts, the above challenges remain inadequately addressed. For challenge ①, some methods (Ashkboos et al., 2024; Li et al., 2024a) alleviates activation outliers using rotation transformation or low-rank decomposition; however, these approaches introduce additional overhead (e.g., QuaRot (Ashkboos et al., 2024) incurs a $0.3\times$ speedup loss on LLaMA2-13B-int4, SVDQuant (Li et al., 2024a) incurs a $0.2\times$ speedup loss on FLUX.1-dev-int4). Moreover, they rely on customized CUDA kernels. By contrast, scaling-based methods (Li et al., 2024c; Shao et al., 2023) address outliers with zero computational overhead. OmniQuant (Shao et al., 2023) optimizes scaling factors via backpropagation, but suffers from training instability and expensive cost (e.g., 7.3 hours of training for LLaMA-30B on an A100-80G). On the other hand, existing training-free scaling methods (Xiao et al., 2023; Wei et al., 2023; Li et al., 2023b) are empirically designed. For example, SmoothQuant (Xiao et al., 2023) aligns the range of activations and weights via per-channel averages. RepQ-ViT (Li et al., 2023b) enforces identical activation ranges across channels. Due to lack of theoretical justification, these methods remain suboptimal and offer no guarantee of effectiveness. For challenge ②, previous methods (Yao et al., 2022; Dettmers et al., 2022) for LLMs address highly dynamic activations by employing dynamic token-wise quantization. Unfortunately, this introduces additional calibration overhead during inference (e.g., LLM.int8 (Dettmers et al., 2022) incurs a $0.5\times$ speedup loss on GPT-3-13B), and the min-max calibration results in accuracy degradation (e.g., a 15.3 FID drop on dynamic token-wise quantization for VAR). ViDiT-Q (Zhao et al., 2024) identifies token-wise variance in Diffusion Transformer (Peebles & Xie, 2023) models, but it adopts the same dynamic strategy as used in LLMs. For challenge ③, current calibration strategies (Shang et al., 2023; Li et al., 2023a; Liu et al., 2024d) focus on temporal-wise mismatch in diffusion models (e.g., EDA-DM (Liu et al., 2024d) extracts samples from various timesteps), yet they fail to address sample-wise mismatch in ARVG models.

To this end, we propose **PTQ4ARVG**, a training-free PTQ framework tailored for ARVG models. **(1)** We conduct an in-depth analysis of scaling effects on quantization and propose Gain-Projected Scaling (GPS), which mitigates channel-wise outliers and represents the first quantization scaling strategy based on mathematical optimization. GPS accurately quantifies the gain of scaling for quantization and derives the optimal scaling factor through mathematical differentiation. Specifically, we perform Taylor expansion on the quantization loss of activations and weights separately. The gain of scaling is defined as the reduction in activation quantization loss minus the increase in weight quantization loss. By differentiating with respect to the scaling factor, we maximize this gain to obtain the optimal

factor that minimizes overall quantization loss. **(2)** We introduce Static Token-Wise Quantization (STWQ) that assigns fine-grained static quantization parameters to handle highly dynamic activations. Since ARVG generates a fixed number of tokens, STWQ allows quantization parameters to be set offline, introducing no online calibration overhead. Moreover, we reveal the position-invariant distributions of token activations across samples, enabling STWQ with a percentile calibration to ensure high accuracy. We also deploy the quantized model to demonstrate the compatibility of STWQ with standard CUDA kernels. **(3)** We design Distribution-Guided Calibration (DGC), which selects samples by evaluating their contribution to the overall distribution entropy. By eliminating redundant samples, DGC enables accurate calibration with a distribution-matched samples. Overall, our contributions are as follows:

- We identify three key challenges in quantizing ARVG models: (1) severe outliers at channel-wise level, (2) highly dynamic activations at token-wise level, and (3) mismatched distribution information at sample-wise level.
- We propose PTQ4ARVG, which includes: (1) GPS leverages mathematical theory to derive optimal scaling factors for outlier suppression, (2) STWQ addresses token-wise variance without incurring additional calibration overhead, and (3) DGC selects samples that match the real distribution to ensure accurate calibration.
- To the best of our knowledge, PTQ4ARVG is the first comprehensive PTQ framework for ARVG family models. We conduct extensive experiments on VAR, RAR, PAR, and MAR, demonstrating that PTQ4ARVG outperforms existing methods and effectively quantizes models to 6-bit while preserving competitive accuracy.

## 2 RELATED WORK

**AutoRegressive Visual Generation models** (Tian et al., 2024; Wang et al., 2024; Yu et al., 2024; Li et al., 2024b) have recently surpassed diffusion models in image generation. More compellingly, their architectural compatibility with LLMs offers great potential for future multimodal integration. As shown in Fig. 2, similar to LLMs, ARVG models rely on an autoregressive transformer architecture to predict the next tokens. However, unlike LLMs with non-fixed token sequence lengths, ARVG predicts a fixed number of tokens. In addition, ARVG enforces bidirectional token dependencies by embedding *conditioning* into the network, which includes positional and class information. Existing ARVG models typically use conditional information as initial token, but differ slightly in token prediction granularity. For example, VAR (Tian et al., 2024) predicts scale tokens at once, RAR (Yu et al., 2024) generates one token at a time, PAR (Wang et al., 2024) first predicts one token sequentially–followed by parallel prediction of multiple non-local tokens, and MAR (Li et al., 2024b) predicts multiple random tokens at once. While current models incorporate KV Cache techniques to accelerate inference, the latency is still unsatisfactory. For instance, PAR-3B takes more than **3** seconds to generate one image. Moreover, the challenge of large model size remains unaddressed. These limitations significantly hinder the deployment and scalability of ARVG models on resource-constrained devices.

**Post-Training Quantization** (Wu et al., 2024; Liu et al., 2024d; Li et al., 2024a; 2023b; Xiao et al., 2023; Shao et al., 2023; Ashkboos et al., 2024; Zhang et al., 2025) reduces model size and accelerates inference. For vision transformer models, RepQ-ViT (Li et al., 2023b) employs reparameterization to address outlier activations. For diffusion models, EDA-DM (Liu et al., 2024d) optimizes the quantization reconstruction loss and introduces a temporally aligned calibration. PTQ4DiT (Wu et al., 2024) adjusts activation and weight distributions based on their correlation in temporal networks. SVDQuant (Li et al., 2024a) isolates outliers via low-rank decomposition and designs customized CUDA kernels to fuse related operations. For LLMs, SmoothQuant (Xiao et al., 2023) performs per-channel equivalent scaling to balance the ranges of activations and weights. OS+ (Wei et al., 2023) further aligns all activation channels to a common center. OmniQuant (Shao et al., 2023) optimizes scaling factors via training, while QuaRot (Ashkboos et al., 2024) alleviates outliers based on rotation transformation. Although these methods perform effectively for previous models, they do not work well with ARVG models. SVDQuant relies on customized CUDA kernels and QuaRot introduces extra inference overhead. EDA-DM and PTQ4DiT are tailored to the temporal nature of diffusion models, making them incompatible with ARVG. In addition, other approaches are inadequate for ARVG. LiteVAR (Xie et al., 2024) pioneers the quantization of VAR models, but it only assigns higher precision to quantization-sensitive layers. Notably, to the best of our knowledge, our method is the first comprehensive PTQ framework specifically designed for ARVG models.

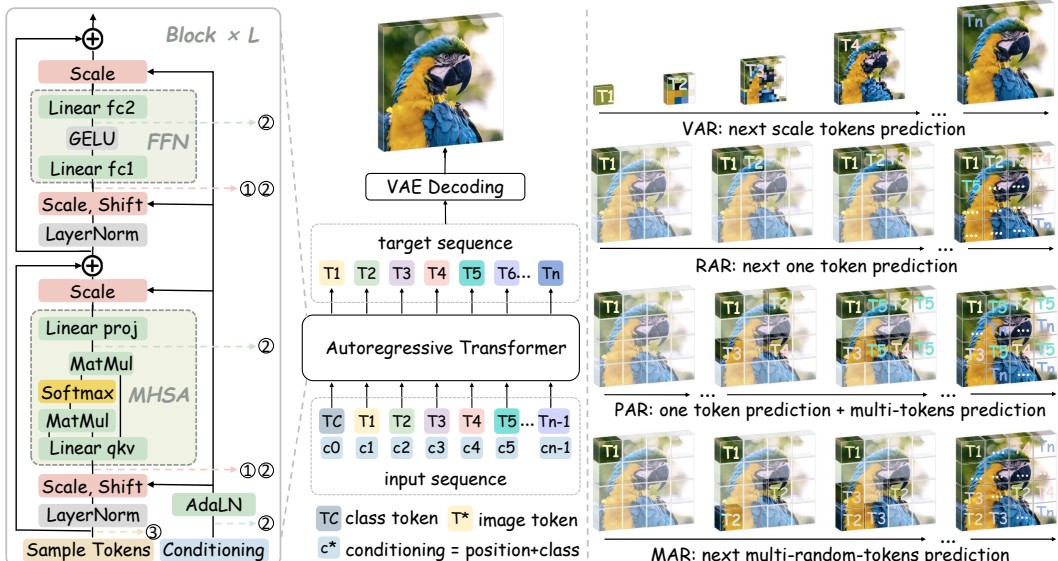

Figure 2: **Overview of ARVG models. (Left)** The autoregressive architecture, mechanism, and challenges of ARVG. **(Right)** Existing ARVG models with different token prediction granularity.

## 3 PRELIMINARY

**ARVG Architecture** consists of $L$ blocks, each containing a Multi-Head Self-Attention (MHSA), a Feedforward Network (FFN), and an Adaptive LayerNorm (AdaLN), as shown in Fig. 2. The AdaLN transforms the *conditioning* into shift and scale parameters to adjust the activation distribution, thereby preserving bidirectional token dependencies and conditional guidance.

**Quantization** transforms a floating-point tensor $x$ to an integer tensor $\bar{x}$ using quantization parameters: scale factor $\delta$ and zero point $z$. The uniform quantization can be formulated as:

$$\bar{x} = \text{clamp}(\lfloor \frac{x}{\delta} \rceil + z, 0, 2^b - 1), \ \ \delta = \frac{R_x}{2^b - 1}, \ \ R_x = x_{up} - x_{down}, \ \ z = \left\lfloor \frac{-x_{down}}{\delta} \right\rceil \quad (1)$$

where $\lfloor \cdot \rceil$ denotes the rounding-to-nearest operator, bit-width $b$ determines the range of clamp function, and $R_x$ represents the quantization range $[x_{up}, x_{down}]$. Min-max calibration calculates the $R_x$ using the minimum and maximum values of $x$. On the other hand, percentile and MSE calibration utilize the percentile values and the minimum quantization error values of $x$, respectively. While the min-max calibration is the fastest, it offers the lowest accuracy. To reduce inference overhead, dynamic token-wise quantization in LLMs typically adopts min-max calibration.

**Equivalent Scaling** is a per-channel transformation that offline shifts the quantization difficulty from activations to weights. For a linear layer with $X \in \mathbb{R}^{T \times n}$ and $W \in \mathbb{R}^{n \times m}$, the output $Y = XW$, $Y \in \mathbb{R}^{T \times m}$, where $T$ is the number of tokens, $n$ is the input channel, and $m$ is the output channel. The activation $X$ divides a per-in-channel scaling factor $s \in \mathbb{R}^n$, and weight $W$ scales accordingly in the reverse direction to maintain mathematical equivalence:

$$Y = (X \oslash s)(s \odot W) \quad (2)$$

Since the $s$ can be fused into the network weights offline, no additional overhead is introduced.

## 4 PTQ4ARVG

### 4.1 GAIN-PROJECTED SCALING

Previous methods primarily focused on distribution-based scaling, relying on empirical intuition to address channel-wise outliers. Due to lack of theoretical justification, these methods remain suboptimal and offer no guarantee of effectiveness. To address the above limitations, we propose **Gain-Projected Scaling (GPS)**, which provides

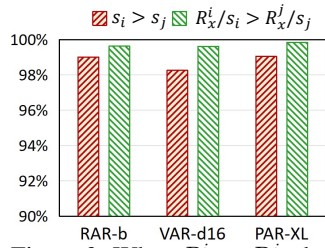

Figure 3: When $R_x^i > R_x^j$, the statistical results of Remark 1.

a mathematical interpretation of how scaling affects quantization, and derives the optimal scaling factor through analytical differentiation:

**Step 1: Quantization Loss.** We begin by analyzing the weight-activation quantization loss $E(\boldsymbol{x}, \boldsymbol{W})$. For a linear layer with activation $\boldsymbol{x} \in \mathbb{R}^{1 \times n}$, weight $\boldsymbol{W} \in \mathbb{R}^{n \times m}$, and output $\boldsymbol{y} \in \mathbb{R}^{1 \times m}$, $\boldsymbol{y} = \boldsymbol{x}\boldsymbol{W}$, the overall quantization loss $E(\boldsymbol{x}, \boldsymbol{W})$ can be formulated as:

$$E(\boldsymbol{x}, \boldsymbol{W}) \leq \mathbb{E}[\mathcal{L}(\hat{\boldsymbol{x}}, \boldsymbol{W}) - \mathcal{L}(\boldsymbol{x}, \boldsymbol{W})] + \mathbb{E}[\mathcal{L}(\boldsymbol{x}, \hat{\boldsymbol{W}}) - \mathcal{L}(\boldsymbol{x}, \boldsymbol{W})] \tag{3}$$

The proof is reported in Appendix A.2. Here, $\hat{\boldsymbol{x}}$ and $\hat{\boldsymbol{W}}$ denote the quantized values. The first and second term in Eq. 3 represent the quantization loss of activation $E_{\boldsymbol{x}}$ and weight $E_{\boldsymbol{W}}$, respectively.

**Step 2: Taylor Expansion.** Similar to prior works AdaRound (Nagel et al., 2020) and BRECQ (Li et al., 2021), we first approximate the weight quantization loss using a Taylor series expansion:

$$E_{\boldsymbol{W}} = \mathbb{E}\left[\mathcal{L}(\boldsymbol{x}, \hat{\boldsymbol{W}}) - \mathcal{L}(\boldsymbol{x}, \boldsymbol{W})\right] \approx \frac{1}{2}\Delta\boldsymbol{W}^T\mathbf{H}^{(\mathbf{W})}\Delta\boldsymbol{W} \tag{4}$$

where $\mathbf{H}^{(\mathbf{W})} = \mathbb{E}\left[\bigtriangledown^2_{\mathbf{W}}\mathcal{L}\right]$ is the Hessian matrix, $\Delta\boldsymbol{W}$ represents the quantization errors of weights. Furthermore, we derive the weight quantization loss for the output $y_k = \boldsymbol{x}\boldsymbol{W}_{:,k}$ as follows:

$$\begin{aligned}E_{\boldsymbol{W}_{:,k}} &\approx \frac{1}{2}\Delta\boldsymbol{W}_{:,k}{}^T\mathbf{H}^{(\mathbf{W}_{:,\mathbf{k}})}\Delta\boldsymbol{W}_{:,k} \\ &\approx \frac{1}{2}\bigtriangledown^2_{\mathbf{y_k}}\mathcal{L} \cdot \mathbb{E}\left[\Delta W_{1,k}{}^2 x_1{}^2 + \Delta W_{2,k}{}^2 x_2{}^2 + ... + \Delta W_{n,k}{}^2 x_n{}^2\right]\end{aligned} \tag{5}$$

Here, $\bigtriangledown^2_{\mathbf{y_k}}\mathcal{L}$ is the Hessian of the task loss w.r.t. $y_k$. We approximate the Hessian loss using the MSE loss, and safely omit the cross terms (please see Appendix A.3 for detail). In the same way, the activation quantization loss for $y_k$ can be formulated as:

$$E_{\boldsymbol{x}} \approx \frac{1}{2}\Delta\boldsymbol{x}^T\mathbf{H}^{(\mathbf{x})}\Delta\boldsymbol{x} \approx \frac{1}{2}\bigtriangledown^2_{\mathbf{y_k}}\mathcal{L} \cdot \mathbb{E}\left[W_{1,k}{}^2\Delta x_1{}^2 + W_{2,k}{}^2\Delta x_2{}^2 + ... + W_{n,k}{}^2\Delta x_n{}^2\right] \tag{6}$$

**Step 3: Introducing Scaling.** To clearly illustrate the impact of scaling on quantization, we simplify the representations of activation $\boldsymbol{x} \in \mathbb{R}^{1 \times 2}$, weight $\boldsymbol{W} \in \mathbb{R}^{2 \times 3}$, output $\boldsymbol{y} \in \mathbb{R}^{1 \times 3}$, and scaling factor $\boldsymbol{s} \in \mathbb{R}^{1 \times 2}$ as follows:

$$\boldsymbol{x} = [\begin{array}{cc} x_1 & x_2 \end{array}], \boldsymbol{W} = \left[\begin{array}{ccc} W_{1,1} & W_{1,2} & W_{1,3} \\ W_{2,1} & W_{2,2} & W_{2,3} \end{array}\right], \boldsymbol{y} = [\begin{array}{ccc} y_1 & y_2 & y_3 \end{array}], \boldsymbol{s} = [\begin{array}{cc} s_1 & s_2 \end{array}] \tag{7}$$

Based on Eq. 5 and Eq. 6, the quantization losses of weight and activation for $y_1$ are:

$$E_{\boldsymbol{W}_{:,1}} \approx \frac{1}{2}\mathbb{E}\left[\Delta W_{1,1}{}^2 x_1{}^2 + \Delta W_{2,1}{}^2 x_2{}^2\right], \quad E_{\boldsymbol{x}} \approx \frac{1}{2}\mathbb{E}\left[W_{1,1}{}^2\Delta x_1{}^2 + W_{2,1}{}^2\Delta x_2{}^2\right] \tag{8}$$

where since $\bigtriangledown^2_{\mathbf{y_1}}\mathcal{L}$ is identical in $E_{\boldsymbol{W}_{:,1}}$ and $E_{\boldsymbol{x}}$, we simplify their expressions accordingly. Furthermore, we represent the quantization range of activations along the channel dimension as $R_{\boldsymbol{x}} = [R_{\boldsymbol{x}}^1, R_{\boldsymbol{x}}^2]$. Since our method adopts a percentile calibration, the range after scaling becomes $R_{\boldsymbol{x}}' = [R_{\boldsymbol{x}}^1/s_1, R_{\boldsymbol{x}}^2/s_2]$. The superscript " $'$ " indicates the values after scaling.

**Analyzing the impact of scaling on quantization.** We quantify the impact of scaling on quantization based on a remark that the activation channel with the largest range before scaling remains the largest range after scaling, formally stated as Remark 1.

**Remark 1.** *When $R_{\boldsymbol{x}}^i > R_{\boldsymbol{x}}^j$, it holds that $s_i > s_j$ and $R_{\boldsymbol{x}}^i/s_i > R_{\boldsymbol{x}}^j/s_j$.*

The remark is based on statistical observations: when $R_{\boldsymbol{x}}^i > R_{\boldsymbol{x}}^j$, over 98% of channels satisfy $s_i > s_j$, and more than 99.5% of channels satisfy $R_{\boldsymbol{x}}^i/s_i > R_{\boldsymbol{x}}^j/s_j$, as shown in Fig. 3.

Without loss of generality, we assume $R_{\boldsymbol{x}}^1 > R_{\boldsymbol{x}}^2$. Inspired by DilateQuant (Liu et al., 2024c), we quantify the quantization error with $\Delta\boldsymbol{x} = \delta_{\boldsymbol{x}} \times c = \frac{R_{\boldsymbol{x}}}{2^b - 1} \times c$, where the $c$ is a integer deviation constant. Based on per-tensor activation quantization and Remark 1, the quantization error before and after scaling can be formulated as:

$$\text{Before}: \Delta x_1 \approx \frac{R_{\boldsymbol{x}}^1}{2^b - 1} \times c_1, \quad \Delta x_2 \approx \frac{R_{\boldsymbol{x}}^1}{2^b - 1} \times c_2 \tag{9}$$

$$\text{After}: \Delta x_1' \approx \frac{R_{\boldsymbol{x}}^1/s_1}{2^b - 1} \times c_1 \approx \Delta x_1/s_1, \quad \Delta x_2' \approx \frac{R_{\boldsymbol{x}}^1/s_1}{2^b - 1} \times c_2 \approx \Delta x_2/s_1 \tag{10}$$

Substituting Eq. 2 and Eq. 10 into Eq. 8, the activation quantization loss after scaling is written as:

$$E'_{\boldsymbol{x}} \approx \frac{1}{2}\mathbb{E}\left[{W'_{1,1}}^2 {\Delta x'_1}^2 + {W'_{2,1}}^2 {\Delta x'_2}^2\right] \approx \frac{1}{2}\mathbb{E}\left[(W_{1,1}\cdot s_1)^2(\Delta x_1/s_1)^2 + (W_{2,1}\cdot s_2)^2(\Delta x_2/s_1)^2\right]$$
$$\approx \frac{1}{2}\mathbb{E}\left[{W_{1,1}}^2 {\Delta x_1}^2 + \frac{{s_2}^2}{{s_1}^2}{W_{2,1}}^2 {\Delta x_2}^2\right] \tag{11}$$

Similarly, as proven in Appendix A.5, the weight quantization loss after scaling can be expressed as:

$$E'_{\boldsymbol{W}_{:,1}} \approx \frac{1}{2}\mathbb{E}\left[{\Delta W_{1,1}}^2 {x_1}^2 + \frac{{s_1}^2}{{s_2}^2}{\Delta W_{2,1}}^2 {x_2}^2\right] \tag{12}$$

In general, $\boldsymbol{s} > 1$ (by $R_{\boldsymbol{x}} > R_{\boldsymbol{W}}$) and $s_1 > s_2$ (by Remark 1), leading to $E'_{\boldsymbol{x}} < E_{\boldsymbol{x}}$ and $E'_{\boldsymbol{W}_{:,1}} > E_{\boldsymbol{W}_{:,1}}$. Therefore, scaling reduces the activation quantization loss, and the gain can be defined as $g_{\boldsymbol{x}} = E_{\boldsymbol{x}} - E'_{\boldsymbol{x}} = \frac{{s_1}^2 - {s_2}^2}{2{s_1}^2}{W_{2,1}}^2 {\Delta x_2}^2$. On the other hand, scaling increases the weight quantization loss, and the added loss can be defined as $g_{\boldsymbol{W}_{:,1}} = E'_{\boldsymbol{W}_{:,1}} - E_{\boldsymbol{W}_{:,1}} = \frac{{s_1}^2 - {s_2}^2}{2{s_2}^2}{\Delta W_{2,1}}^2 {x_2}^2$.

**Step 4: Closed-form solution of scaling factor.** We model a scaling gain function to derive the optimal scaling factor:

$$g(s_2) = g_{\boldsymbol{x}} - g_{\boldsymbol{W}_{:,1}} = \frac{1}{2}\left({W_{2,1}}^2 {\Delta x_2}^2 + {\Delta W_{2,1}}^2 {x_2}^2 - \frac{{s_2}^2}{{s_1}^2}{W_{2,1}}^2 {\Delta x_2}^2 - \frac{{s_1}^2}{{s_2}^2}{\Delta W_{2,1}}^2 {x_2}^2\right) \tag{13}$$

Here, $s_1$ denotes the scaling factor for the channel with the largest activation range. We calculate it using $s_1 = \sqrt{R_x^1/R_W^1}$, which ensures that ${R_x^1}' = {R_W^1}'$. In conclusion, **we reformulate the problem of determining all scaling factors into solving the remaining scaling factors based on** $s_1$. Specifically, $s_2$ is optimized with respect to $s_1$ to maximize $g(s_2)$. Clearly, this is a convex optimization problem, which can be solved by taking the derivative:

$$g'(s_2) = -\frac{s_2}{{s_1}^2}{W_{2,1}}^2 {\Delta x_2}^2 + \frac{{s_1}^2}{{s_2}^3}{\Delta W_{2,1}}^2 {x_2}^2 \Rightarrow s_2 = s_1\frac{\sqrt{|\Delta W_{2,1}x_2|}}{\sqrt{|W_{2,1}\Delta x_2|}} \tag{14}$$

The above derivation only minimizes the quantization loss of $y_1$. We further extend it to $\boldsymbol{y}$ yields:

$$g(s_2) = \frac{1}{2}\sum_{i=1}^{m}({W_{2,i}}^2 {\Delta x_2}^2 + {\Delta W_{2,i}}^2 {x_2}^2) - \frac{{s_2}^2}{2{s_1}^2}\sum_{i=1}^{m}{W_{2,i}}^2 {\Delta x_2}^2 - \frac{{s_1}^2}{2{s_2}^2}\sum_{i=1}^{m}{\Delta W_{2,i}}^2 {x_2}^2 \tag{15}$$

$$s_2 = s_1\frac{\sqrt{\sum_{i=1}^{m}|\Delta W_{2,i}x_2|}}{\sqrt{\sum_{i=1}^{m}|W_{2,i}\Delta x_2|}} \tag{16}$$

Based on Eq. 16, we maximize scaling gain to obtain the optimal scaling factor that minimizes overall quantization loss. The Algorithm of GPS is illustrated in Appendix F. It is worth noting that different $\nabla^2_{\boldsymbol{y_k}}\mathcal{L}$ are unobservable and exhibit slight variations. Fortunately, prior optimization studies (Nagel et al., 2020; Li et al., 2021; Wei et al., 2022) assume them to be a common constant, which does not affect the optimization results. We follow the same assumption in our method.

## 4.2 STATIC TOKEN-WISE QUANTIZATION

We reveal that ARVG models exhibit highly dynamic activations at token-wise level, characterized by: (1) input of the AdaLN module showing variation along the token dimension (Fig. 1(b)). (2) input of linear layers existing *sink* tokens (Fig. 1(c)). This phenomenon is further analyzed in Appendix H. LLMs also exhibit token-wise activation variation. However, due to the variable token sequence lengths and position-uncertain token distributions, only dynamic token-wise quantization (Shao et al., 2023; Dettmers et al., 2022) with online min-max calibration can be applied. This method not only introduces additional overhead during inference but also leads to inaccurate calibration.

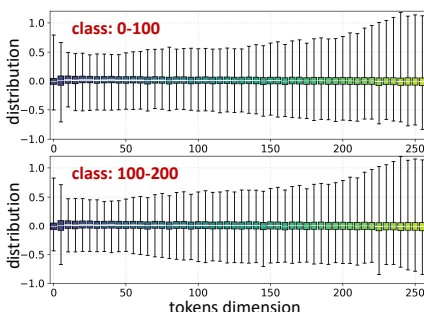

Figure 4: Inputs of AdaLN in RAR-B from different sample class. The distribution remains invariant across samples.

In sharp contrast, ARVG models exhibit two distinctive properties: **fixed token sequence lengths** and **position-invariant distribution across samples** (as shown in Fig. 4). Building on these properties, we propose **Static Token-Wise Quantization (STWQ)**, which sets quantization parameters by offline percentile calibration. Specifically, as shown in Appendix Fig. 9, (1) STWQ assigns quantization parameters along the token sequence for AdaLN module. (2) STWQ separately assigns quantization parameters for the *sink* tokens and normal tokens of linear layers. As reported in Table 5, STWQ introduces no additional calibration overhead while preserving accuracy.

## 4.3 DISTRIBUTION-GUIDED CALIBRATION

Current calibration research focuses on temporal-wise redundancy in diffusion models. DiTFastAttn (Yuan et al., 2024) and LiteVAR (Xie et al., 2024) are the first to recognize sample-wise redundancy. They leverages this property for model caching. However, we find that the sample-wise redundancy also results in mismatched calibration, significantly hindering accurate quantization. To address this, we propose **Distribution-Guided Calibration (DGC)**, which employs Mahalanobis distance to measure the distributional entropy $\rho$ of a sample $x$ with respect to a sample set:

$$\rho(x) = \sqrt{(x - u)^T S^{-1} (x - u)} \tag{17}$$

where $u$ and $S$ represent the mean and covariance of the sample set, respectively. DGC selects the top 50% of samples with the highest distributional entropy as the calibration.

## 5 EXPERIMENT

**Models and Metrics.** We assess the quantization capabilities of our method on four ARVG models: VAR (Tian et al., 2024), RAR (Yu et al., 2024), PAR (Wang et al., 2024), and MAR (Li et al., 2024b). All models generate 50K images on ImageNet (Deng et al., 2009) and are evaluated using FID (Heusel et al., 2017), sFID, IS (Salimans et al., 2016), and Precision. We also deploy the quantized models on an RTX 3090 GPU to assess real-world acceleration and compression performance.

Table 1: Comparative results for VAR and RAR models.

| Bit Width | Methods | VAR-d16 | | | | VAR-d24 | | | |
|---|---|---|---|---|---|---|---|---|---|
| | | IS ↑ | FID ↓ | sFID ↓ | Precision ↑ | IS ↑ | FID ↓ | sFID ↓ | Precision ↑ |
| FP | - | 283.21 | 3.60 | 8.27 | 0.85 | 317.16 | 2.33 | 8.24 | 0.82 |
| W8A8 | SmoothQuant | 229.87 | 4.29 | 13.39 | 0.79 | 246.68 | 4.42 | 12.66 | 0.77 |
| | RepQ* | 211.21 | 4.36 | 13.33 | 0.76 | 240.18 | 4.74 | 14.10 | 0.76 |
| | OS+ | 230.41 | 4.11 | 12.22 | 0.79 | 250.61 | 4.14 | 12.93 | 0.77 |
| | OmniQuant | 226.92 | 4.19 | 12.49 | 0.79 | 244.46 | 5.20 | 14.98 | 0.76 |
| | QuaRot | 231.38 | 3.99 | 11.38 | 0.79 | 257.71 | 3.40 | 13.34 | 0.77 |
| | SVDQuant | 229.36 | 4.11 | 12.72 | 0.78 | 253.78 | 3.29 | 12.38 | 0.76 |
| | Ours | 230.04 | 4.06 | 12.23 | 0.79 | 252.70 | 3.36 | 13.24 | 0.77 |
| W6A6 | SmoothQuant | 101.55 | 18.54 | 17.22 | 0.57 | 178.43 | 7.93 | 12.80 | 0.65 |
| | RepQ* | 109.97 | 16.30 | 15.57 | 0.59 | 160.65 | 8.84 | 15.45 | 0.66 |
| | OS+ | 123.38 | 13.54 | 12.68 | 0.64 | 191.61 | 6.54 | 13.40 | 0.67 |
| | OmniQuant | 98.27 | 22.19 | 19.44 | 0.57 | 115.02 | 18.35 | 23.40 | 0.61 |
| | QuaRot | 155.76 | 8.96 | 13.26 | 0.67 | 200.58 | 5.90 | 13.68 | 0.70 |
| | SVDQuant | 130.15 | 12.53 | 15.36 | 0.60 | 195.83 | 6.23 | 13.12 | 0.70 |
| | Ours | 162.85 | 8.34 | 12.63 | 0.68 | 204.02 | 5.51 | 12.73 | 0.71 |

| Bit Width | Methods | RAR-B | | | | RAR-XL | | | |
|---|---|---|---|---|---|---|---|---|---|
| FP | - | 292.80 | 1.96 | 6.16 | 0.82 | 308.54 | 1.54 | 5.31 | 0.80 |
| W8A8 | SmoothQuant | 242.97 | 2.80 | 7.76 | 0.78 | 229.06 | 3.35 | 8.33 | 0.74 |
| | RepQ* | 211.64 | 4.37 | 9.70 | 0.73 | 212.13 | 4.22 | 8.89 | 0.71 |
| | OS+ | 256.01 | 2.50 | 7.93 | 0.78 | 238.01 | 3.13 | 8.53 | 0.73 |
| | OmniQuant | 281.12 | 2.23 | 7.21 | 0.81 | 283.67 | 1.68 | 5.92 | 0.78 |
| | QuaRot | 250.54 | 3.41 | 13.07 | 0.77 | 279.09 | 2.06 | 6.47 | 0.79 |
| | SVDQuant | 164.69 | 11.94 | 22.09 | 0.69 | 279.94 | 1.86 | 5.85 | 0.76 |
| | Ours | 283.47 | 2.21 | 7.22 | 0.82 | 304.18 | 1.58 | 5.57 | 0.80 |
| W6A6 | SmoothQuant | 31.04 | 63.77 | 42.08 | 0.36 | 30.00 | 63.70 | 40.84 | 0.36 |
| | RepQ* | 18.96 | 82.31 | 49.57 | 0.43 | 22.50 | 77.11 | 44.70 | 0.30 |
| | OS+ | 57.92 | 40.14 | 24.58 | 0.45 | 19.82 | 84.63 | 54.38 | 0.30 |
| | OmniQuant | 148.89 | 11.66 | 12.89 | 0.67 | 150.35 | 13.31 | 13.23 | 0.64 |
| | QuaRot | 18.86 | 101.60 | 38.43 | 0.32 | 43.32 | 54.40 | 40.40 | 0.42 |
| | SVDQuant | 8.97 | 125.51 | 65.60 | 0.39 | 200.80 | 5.41 | 7.75 | 0.68 |
| | Ours | 206.17 | 5.13 | 12.68 | 0.75 | 250.70 | 2.79 | 6.70 | 0.76 |

Table 2: Comparative results for PAR models.

| Bit Width | Methods | PAR-XL-4× | | | | PAR-XXL-4× | | | |
|---|---|---|---|---|---|---|---|---|---|
| | | IS ↑ | FID ↓ | sFID ↓ | Precision ↑ | IS ↑ | FID ↓ | sFID ↓ | Precision ↑ |
| FP | - | 259.2 | 2.61 | - | 0.82 | 263.2 | 2.35 | - | 0.82 |
| W8A8 | SmoothQuant | 8.28 | 132.17 | 79.50 | 0.06 | 4.35 | 207.84 | 152.22 | 0.15 |
| | RepQ* | 6.44 | 138.97 | 114.35 | 0.06 | 5.90 | 188.37 | 101.15 | 0.14 |
| | OS+ | 7.31 | 128.72 | 93.63 | 0.07 | 4.89 | 192.82 | 125.03 | 0.14 |
| | OmniQuant | 215.71 | 3.55 | 7.78 | 0.78 | 224.87 | 3.05 | 6.73 | 0.78 |
| | SVDQuant | 213.98 | 3.80 | 7.79 | 0.78 | 222.74 | 2.91 | 6.83 | 0.78 |
| | Ours | 219.50 | 3.55 | 7.71 | 0.79 | 232.16 | 2.95 | 6.60 | 0.79 |
| W6A6 | SmoothQuant | 8.08 | 146.52 | 69.36 | 0.05 | 4.90 | 175.21 | 131.43 | 0.17 |
| | RepQ* | 5.82 | 157.07 | 132.66 | 0.05 | 4.04 | 210.32 | 139.97 | 0.12 |
| | OS+ | 7.27 | 138.78 | 86.01 | 0.06 | 4.05 | 194.58 | 144.41 | 0.22 |
| | OmniQuant | 15.72 | 107.29 | 53.01 | 0.20 | 15.15 | 113.72 | 73.68 | 0.18 |
| | SVDQuant | 102.03 | 19.52 | 18.19 | 0.60 | 101.71 | 18.80 | 20.56 | 0.63 |
| | SQ+STWQ | 107.33 | 13.26 | 7.09 | 0.59 | 94.27 | 14.69 | 7.73 | 0.59 |
| | RepQ*+STWQ | 72.96 | 23.04 | 9.93 | 0.50 | 82.35 | 20.74 | 12.96 | 0.54 |
| | Ours | 113.33 | 12.87 | 6.80 | 0.62 | 119.19 | 11.05 | 7.17 | 0.63 |

Table 3: Comparative results for MAR models.

| Bit Width | Methods | MAR-B | | | MAR-L | | | MAR-H | | |
|---|---|---|---|---|---|---|---|---|---|---|
| | | IS ↑ | FID ↓ | Precision ↑ | IS ↑ | FID ↓ | Precision ↑ | IS ↑ | FID ↓ | Precision ↑ |
| FP | - | 281.7 | 2.31 | 0.82 | 296.0 | 1.78 | 0.81 | 303.7 | 1.55 | 0.81 |
| W8A8 | OS+ | 169.43 | 5.94 | 0.69 | 224.64 | 2.87 | 0.73 | 248.27 | 2.84 | 0.73 |
| | QuaRot | 173.55 | 6.82 | 0.70 | 178.99 | 7.57 | 0.68 | 175.23 | 9.07 | 0.68 |
| | SVDQuant | 165.47 | 6.40 | 0.69 | 190.18 | 5.19 | 0.68 | 233.13 | 4.21 | 0.71 |
| | Ours | 279.97 | 2.36 | 0.82 | 284.02 | 1.92 | 0.80 | 294.66 | 1.67 | 0.79 |
| W6A6 | OS+ | 109.83 | 13.56 | 0.61 | 141.75 | 11.02 | 0.62 | 174.49 | 8.44 | 0.65 |
| | QuaRot | 140.71 | 11.97 | 0.66 | 137.59 | 13.85 | 0.63 | 122.76 | 17.79 | 0.62 |
| | SVDQuant | 51.95 | 36.14 | 0.46 | 30.19 | 68.52 | 0.33 | 10.47 | 142.37 | 0.18 |
| | Ours | 249.14 | 2.99 | 0.78 | 254.13 | 3.12 | 0.76 | 261.35 | 2.62 | 0.75 |

**Quantization and Comparison Settings.** PTQ4ARVG applies 6-bit (W6A6) or 8-bit (W8A8) quantization to all linear layers and matrix multiplications in ARVG models. To highlight the efficiency, PTQ4ARVG only selects 128 samples for calibration. Additionally, our method introduces no additional overhead during inference and does not rely on customized CUDA kernel designs. Since no dedicated quantization framework exists for ARVG models, we compare it with representative quantization approaches, including SmoothQuant (Xiao et al., 2023), OS+ (Wei et al., 2023), RepQ* (Li et al., 2023b), OmniQuant (Shao et al., 2023), Quarot (Ashkboos et al., 2024), and SVDQuant (Li et al., 2024a). Notably, all methods are evaluated under their default settings.

## 5.1 MAIN RESULTS

**VAR and RAR Model.** As reported in Table 1, our method significantly outperforms both training-free OS+ and training-based OmniQuant, improving FID by 35.01 and 6.53 on 6-bit RAR-B, respectively. Compared to the rotation-based QuaRot, our method demonstrates greater advantages at lower bit-widths. Moreover, due to the distinct activation distributions and autoregressive architecture of ARVG, SVDQuant cannot retain the advantages it demonstrates on diffusion models.

**PAR and MAR Model.** As reported in Table 2 and Table 3, the OmniQuant and SVDQuant fail at 6-bit precision, while our method maintains competitive performance. Notably, we do not compare against QuaRot, as PAR does not meet its requirements: " The formula holds when the number of heads and the dimension of each head are both powers of 2 ". More results, including the larger models and W4A8 tasks, are provided in Appendix J.

## 5.2 ABLATION STUDY

We conduct ablation studies using SmoothQuant as the baseline. As reported in Table 4, our method consistently improves quantization performance, demonstrating the effectiveness of each proposed component. In the following, we perform a more fine-grained analysis of each method individually.

**Ablation Study on GPS.** We compare GPS with distribution-based scaling methods on RAR-B at 6-bit precision. Specifically, using PTQ4ARVG as the baseline, we replace GPS with different scaling methods, as reported in Table 6. Here, SQ+RepQ* denotes first applying SmoothQuant, followed by RepQ*. As can be seen, our mathematically-derived scaling method outperforms previous approaches. Furthermore, to

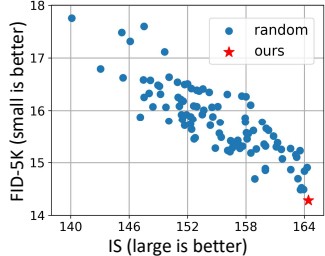

Figure 5: Visualizing the Advantages of GPS on VAR-d16 at 6-bit precision.

Table 4: Efficacy of different component in this paper.

| Method | | | Bit Width | RAR-B | | | | VAR-d16 | | | |
|---|---|---|---|---|---|---|---|---|---|---|---|
| GPS | STWQ | DGC | | IS ↑ | FID ↓ | sFID ↓ | Precision ↑ | IS ↑ | FID ↓ | sFID ↓ | Precision ↑ |
| ✗ | ✗ | ✗ | | 31.04 | 63.77 | 72.08 | 0.36 | 101.55 | 18.54 | 17.22 | 0.57 |
| ✓ | ✗ | ✗ | W6A6 | 62.47 | 36.51 | 24.53 | 0.46 | 127.13 | 13.32 | 14.41 | 0.64 |
| ✓ | ✓ | ✗ | | 183.21 | 6.67 | 12.74 | 0.71 | 161.36 | 8.75 | 13.53 | 0.69 |
| ✓ | ✓ | ✓ | | 206.17 | 5.13 | 12.68 | 0.75 | 162.85 | 8.34 | 12.63 | 0.69 |

validate the optimality of our scaling factor $s_{GPS}$, we introduce random perturbations within the range $[-0.3s_{GPS}, +0.3s_{GPS}]$ to $s_{GPS}$, and conduct a 100 times experiments. As shown in Fig. 5, $s_{GPS}$ achieves the best quantization performance.

Table 5: Ablation experiments of STWQ.

| Method | IS↑ | FID↓ | Precision↑ | Time (ms) | Speedup |
|---|---|---|---|---|---|
| FP | 283.21 | 3.60 | 0.85 | 1163.0 | 1.000× |
| SQ (w/o TW) | 101.55 | 18.54 | 0.57 | 397.1 | 2.929× |
| SQ+DTWQ | 73.05 | 30.14 | 0.49 | 473.9 | 2.457× |
| SQ+STWQ | 151.60 | 10.41 | 0.67 | 397.9 | 2.922× |

Table 6: Ablation experiments of GPS.

| Scaling Method | IS ↑ | FID ↓ | Precision ↑ |
|---|---|---|---|
| SmoothQuant | 135.40 | 10.26 | 0.68 |
| RepQ* | 92.44 | 33.79 | 0.59 |
| OS+ | 161.63 | 7.71 | 0.69 |
| SQ+RepQ* | 170.07 | 7.43 | 0.70 |
| GPS (ours) | 206.17 | 5.13 | 0.75 |

**Ablation Study on STWQ.** We compare STWQ with dynamic token-wise quantization (DTWQ) on 6-bit VAR-d16. As reported in Table 5, STWQ outperforms DTWQ in accuracy. Additionally, we deploy the quantized network (8-bit). With a batch size of 100 and a sequence length of 256, DTWQ results in a 0.47× reduction in speedup compared to the case without token-wise quantization. In contrast, STWQ maintains quantization efficiency while preserving high accuracy. The slight differences in speedup arise from operator scheduling rather than dynamic calibration.

**Ablation Study on DGC.** Previous calibration for diffusion models focused on addressing temporal-wise mismatch. In contrast, DGC, from a novel perspective, eliminates sample-wise mismatch to improve performance. We compare our method with random and uniform sampling methods, as shown in Fig. 6.

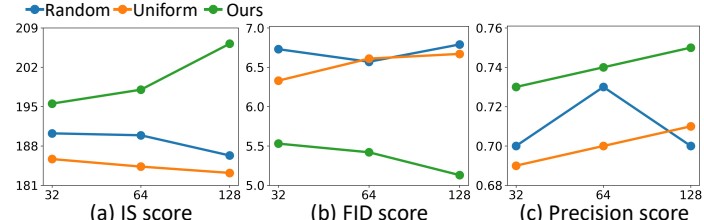

Figure 6: Ablation experiments of DGC on RAR-B with 6-bit quantization. The x-axis denotes different calibration size.

DGC not only achieves high accuracy but also maintains strong robustness, as evidenced by consistent improvements across all metrics with larger calibration sizes.

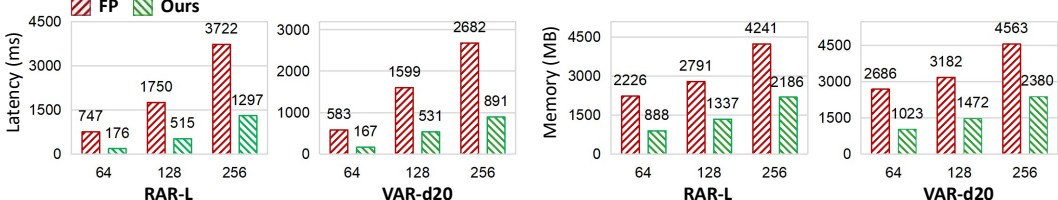

Figure 7: The PyTorch implementation of PTQ4ARVG on RTX 3090 GPU.

**Speedup and Memory Saving.** In this section, we deploy the 8-bit RAR-L and VAR-d20 to evaluate the real acceleration and compression performance. We use a standard CUDA kernel to deploy the decoder network. Inference latency and peak memory usage are evaluated with a batch size of 100 across varying token sequence lengths. As shown in Fig. 7, PTQ4ARVG achieves a 3.01× speedup and a 1.92× reduction in peak memory on VAR-d20 when the sequence length is 256.

## 6 CONCLUSION

In this paper, we focus on introducing the quantization techniques into the realm of ARVG models. Our study reveals the quantization challenges of ARVG models across the channel, token, and sample dimensions. Correspondingly, we propose PTQ4ARVG, a training-free and hardware-friendly PTQ framework tailored for ARVG models. Specifically, (1) we propose a novel theory-based scaling strategy that quantifies scaling gains in quantization and derives the optimal scaling factor via differentiation to address channel outliers. (2) Leveraging ARVG properties, we offline-assign finer-grained quantization parameters to handle highly dynamic token activations. (3) We eliminate redundant samples based on distribution entropy to obtain a distribution-matched calibration. Experiments show that PTQ4ARVG advances in accuracy for ARVG family compared to existing methods, making ARVG models more practical for deployment in resource-constrained environments. We hope our work will further advance the research and applicability of ARVG models.

## 7 ACKNOWLEDGE

This work is supported in part by the Strategic Priority Research Program of Chinese Academy of Sciences under Grant Number XDB1100000; in part by the National Natural Science Foundation of China under Grant Number 62276255; in part by the Postdoctoral Fellowship Program of CPSF under Grant Number GZC20251175.

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

# PTQ4ARVG: Supplementary Materials

## A  PROOF

### A.1  QDrop's Proof of $E(x, W)$

The quantization-dequantization process of a activation $x$ can be represented as:

$$Quant : \bar{x} = \text{clamp}\left(\left\lceil \frac{x}{\delta} \right\rceil + z\right), \quad DeQuant : \hat{x} = \delta \cdot (\bar{x} - z) \approx x \tag{18}$$

where $\bar{x}$ denotes the integer value. The introduction of quantization error to $x$ can be expressed as $\hat{x} = x(1 + u(x))$, where $u$ can be defined as:

$$u = \frac{\hat{x}}{x} - 1 = \frac{(\bar{x} - z) \cdot \delta}{(\bar{x} - z + c) \cdot \delta} - 1 = \frac{\bar{x} - z}{\bar{x} - z + c} - 1 = \frac{-c}{\bar{x} - z + c} \tag{19}$$

here, $c$ denotes the deviation of the integer value, which is affected by bit-width and rounding error, and can thus be treated as a constant.

Furthermore, Consider matrix-vector multiplication, The quantized output can be expressed as $\hat{y} = \hat{x}\hat{W} = x\left(\hat{W} \odot (1 + V(x))\right)$, given by

$$\hat{x}\hat{W} = \left(x \odot \begin{bmatrix} 1 + u_1(x) \\ 1 + u_2(x) \\ \cdots \\ 1 + u_n(x) \end{bmatrix}\right)\hat{W} = x\left(\hat{W} \odot \begin{bmatrix} 1 + u_1(x) & \cdots & 1 + u_n(x) \\ 1 + u_1(x) & \cdots & 1 + u_n(x) \\ & \cdots & \\ 1 + u_1(x) & \cdots & 1 + u_n(x) \end{bmatrix}\right) \tag{20}$$

As can be seen, by taking $V_{i,j}(x) = u_j(x)$, quantization error on the activation vector $(1 + u(x))$ can be transplanted into perturbation on weight $(1 + V(x))$. Thus, the error caused by weight-activation quantization can be briefly expressed as:

$$E(x, W) = \mathbb{E}\left[\mathcal{L}(\hat{x}, \hat{W}) - \mathcal{L}(x, W)\right] \tag{21}$$

$$= \mathbb{E}\left[L\left(x(1 + u(x)), \hat{W}\right) - L(x, W)\right] \tag{22}$$

$$= \mathbb{E}\left[L\left(x, \hat{W} \odot (1 + V(x))\right) - L(x, W)\right] \tag{23}$$

### A.2  Proof of $E(x, W)$

The $E(x, W)$ can be formulated as:

$$E(x, W) = \mathbb{E}\left[\mathcal{L}(\hat{x}, \hat{W}) - \mathcal{L}(x, W)\right] = \mathbb{E}\left[\mathcal{L}(x + \Delta x, W + \Delta W) - \mathcal{L}(x, W)\right] \tag{24}$$

Here, $\hat{x}$ and $\hat{W}$ denote the quantized values, $\Delta x$ and $\Delta W$ represent the quantization errors of the activations and weights, respectively. As proven in QDrop (Wei et al., 2022), $E(x, W)$ admits the form:

$$E(x, W) = \mathbb{E}\left[L\left(x, \hat{W} \odot (1 + V(x))\right) - L(x, W)\right] \tag{25}$$

For convenience, we reproduce QDrop's derivation in Appendix A.1. Evidently, the activation quantization error can be accumulated into the weight quantization error. Therefore, the intermediate state $\mathcal{L}(x, \hat{W})$ can be inserted into the Eq. 25:

$$E(x, W) = \mathbb{E}\left[\mathcal{L}\left(x, \hat{W} \odot (1 + V(x))\right) - \mathcal{L}(x, \hat{W}) + \mathcal{L}(x, \hat{W}) - \mathcal{L}(x, W)\right]$$

$$\leq \mathbb{E}\left[\mathcal{L}\left(x, \hat{W} \odot (1 + V(x))\right) - \mathcal{L}(x, \hat{W})\right] + \mathbb{E}\left[\mathcal{L}(x, \hat{W}) - \mathcal{L}(x, W)\right] \tag{26}$$

$$\leq \mathbb{E}\left[\mathcal{L}(\hat{x}, \hat{W}) - \mathcal{L}(x, \hat{W})\right] + \mathbb{E}\left[\mathcal{L}(x, \hat{W}) - \mathcal{L}(x, W)\right]$$

At this point, the $E(x, W)$ is decomposed into the activation quantization loss $E_x$ (the first term in Eq. 26) and the weight quantization loss $E_W$ (the second term in Eq. 26).

Extensive prior studies (Xiao et al., 2023; Liu et al., 2024c; Li et al., 2023b) have shown that, compared to activation quantization error, the weight quantization error is minimal ($W \approx \hat{W}$) under the weight-activation quantization. **To simplify the derivation**, we approximate the activation quantization loss, $\hat{E}_x = \mathbb{E}\left[\mathcal{L}(\hat{x}, \hat{W}) - \mathcal{L}(x, \hat{W})\right]$, using activation error with the full-precision weights, $E_x = \mathbb{E}[\mathcal{L}(\hat{x}, W) - \mathcal{L}(x, W)]$. The rationale for this approximation is given in Appendix A.4. **Note that in our code implementation we still compute the activation quantization loss via the $\hat{E}_x$.** Finally, the quantization loss $E(x, W)$ is formulated as follows:

$$E(x, W) \leq \mathbb{E}\left[\mathcal{L}(\hat{x}, W) - \mathcal{L}(x, W)\right] + \mathbb{E}\left[\mathcal{L}(x, \hat{W}) - \mathcal{L}(x, W)\right] \tag{27}$$

### A.3 Proof of $E_{W_{:,k}}$

Inspired by AdaRound (Nagel et al., 2020), we approximate the weight quantization loss using the mean-squared error (MSE) loss, as formulated below:

$$E_{W_{:,k}} \approx \frac{1}{2}\Delta W_{:,k}{}^T \mathbf{H}^{(\mathbf{W}_{:,k})}\Delta W_{:,k} \tag{28}$$

$$\approx \frac{1}{2}\mathbb{E}\left[\nabla^2_{\mathbf{y_k}}\mathcal{L} \cdot \Delta W_{:,k}{}^T x^T x \Delta W_{:,k}\right] \tag{29}$$

$$\approx \frac{1}{2}\nabla^2_{\mathbf{y_k}}\mathcal{L} \cdot \Delta W_{:,k}{}^T \mathbb{E}\left[x^T x\right]\Delta W_{:,k} \tag{30}$$

$$\approx \frac{1}{2}\nabla^2_{\mathbf{y_k}}\mathcal{L} \cdot \mathbb{E}[(x\Delta W_{:,k})^2] \tag{31}$$

where $\nabla^2_{\mathbf{y_k}}\mathcal{L}$ is the Hessian of the task loss w.r.t. $y_k$. We further demonstrate that the cross terms are negligible and can therefore be safely omitted, as justified by the following analysis:

$$
\begin{aligned}
E_{W_{:,k}} &\approx \frac{1}{2}\nabla^2_{\mathbf{y_k}}\mathcal{L} \cdot \mathbb{E}[(x\Delta W_{:,k})^2] \\
&\approx \frac{1}{2}\nabla^2_{\mathbf{y_k}}\mathcal{L} \cdot \mathbb{E}\left[(x\Delta W_{:,k}) \cdot (x\Delta W_{:,k})\right] \\
&\stackrel{(a)}{\approx} \frac{1}{2}\nabla^2_{\mathbf{y_k}}\mathcal{L} \cdot \mathbb{E}\left[(\Delta W_{:,k}{}^T x^T) \cdot (x\Delta W_{:,k})\right] \\
&\approx \frac{1}{2}\nabla^2_{\mathbf{y_k}}\mathcal{L} \cdot \mathbb{E}\left[\Delta W_{:,k}{}^T (x^T x)\Delta W_{:,k}\right] \\
&\approx \frac{1}{2}\nabla^2_{\mathbf{y_k}}\mathcal{L} \cdot \mathbb{E}\left[\sum_{i=1}^{n}\sum_{j=1}^{n}\Delta W_{i,k}\Delta W_{j,k}x_i x_j\right] \\
&\approx \frac{1}{2}\nabla^2_{\mathbf{y_k}}\mathcal{L} \cdot \sum_{i=1}^{n}\sum_{j=1}^{n}\mathbb{E}\left[\Delta W_{i,k}\Delta W_{j,k}\right]x_i x_j \\
&\stackrel{(b)}{\approx} \frac{1}{2}\nabla^2_{\mathbf{y_k}}\mathcal{L} \cdot \mathbb{E}\left[\Delta W_{1,k}{}^2 x_1{}^2 + \Delta W_{2,k}{}^2 x_2{}^2 + ... + \Delta W_{n,k}{}^2 x_n{}^2\right]
\end{aligned}
\tag{32}
$$

where (a) since $x\Delta W_{:,k} \in \mathbb{R}^1$, its transpose equals itself; (b) due to the inherent randomness of rounding-to-nearest quantization, each $\Delta W_{i,k}, \Delta W_{j,k} \in \Delta W_{:,k}$ is independent and satisfies $\mathbb{E}[\Delta W_{i,k}\Delta W_{j,k}] = \mathbb{E}[\Delta W_{i,k}]\mathbb{E}[\Delta W_{j,k}] = 0$ when $i \neq j$. Therefore, for $i \neq j$, i.e., in the case of cross terms, these terms can be safely omitted.

### A.4 Proof of $E_x \approx \hat{E}_x$

According to Eq. 6, $E_x$ and $\hat{E}_x$ can be expressed as:

$$E_x = \mathbb{E}\left[\mathcal{L}(\hat{x}, W) - \mathcal{L}(x, W)\right] \tag{33}$$

$$\approx \frac{1}{2}\nabla^2_{\mathbf{y}}\mathcal{L} \cdot \mathbb{E}\left[W_{1,k}{}^2\Delta x_1{}^2 + W_{2,k}{}^2\Delta x_2{}^2 + ... + W_{n,k}{}^2\Delta x_n{}^2\right] \tag{34}$$

$$\hat{E}_{\boldsymbol{x}} = \mathbb{E}\left[\mathcal{L}(\hat{\boldsymbol{x}}, \hat{\boldsymbol{W}}) - \mathcal{L}(\boldsymbol{x}, \hat{\boldsymbol{W}})\right] \tag{35}$$

$$\approx \frac{1}{2}\nabla_{\mathbf{y}}^2 \mathcal{L} \cdot \mathbb{E}\left[\hat{W}_{1,k}^2 \Delta x_1^2 + \hat{W}_{2,k}^2 \Delta x_2^2 + \cdots + \hat{W}_{n,k}^2 \Delta x_n^2\right]$$

$$\approx \frac{1}{2}\nabla_{\mathbf{y}}^2 \mathcal{L} \cdot \mathbb{E}\Big[\big(W_{1,k}{}^2 + 2W_{1,k}\Delta W_{1,k} + \Delta W_{1,k}{}^2\big)\Delta x_1^2$$

$$+ \big(W_{2,k}{}^2 + 2W_{2,k}\Delta W_{2,k} + \Delta W_{2,k}{}^2\big)\Delta x_2^2$$

$$+ \dots$$

$$+ \big(W_{n,k}{}^2 + 2W_{n,k}\Delta W_{n,k} + \Delta W_{n,k}{}^2\big)\Delta x_n^2\Big] \tag{36}$$

Since $\Delta \boldsymbol{W}$ is small and $\mathbb{E}[\Delta \boldsymbol{W}] = 0$, we neglect the higher-order terms in $\hat{E}_{\boldsymbol{x}}$. Therefore, $E_{\boldsymbol{x}}$ can be numerically approximated by $\hat{E}_{\boldsymbol{x}}$.

### A.5 Proof of $E'_{\boldsymbol{W}_{:,1}}$

Since weights are quantized per output channel, The quantization range of weight $\boldsymbol{W}_{:,1}$ is denoted as $R_W^1$. We also observe that: *when $s_i > s_j$, the new quantization range $R_W^1{}'$ is better approximated by $R_W^1 \cdot s_i$ (as shown in Fig. 8).* So the quantization error for weight before and after scaling can be expressed as:

$$\text{Before}: \ \Delta W_{1,1} \approx \frac{R_W^1}{2^b - 1} \times c_3, \quad \Delta W_{2,1} \approx \frac{R_W^1}{2^b - 1} \times c_4 \tag{37}$$

$$\text{After}: \ \Delta W_{1,1}{}' \approx \frac{R_W^1 \cdot s_1}{2^b - 1} \times c_3 \approx \Delta W_{1,1} \cdot s_1, \Delta W_{2,1}{}' \approx \frac{R_W^1 \cdot s_1}{2^b - 1} \times c_4 \approx \Delta W_{2,1} \cdot s_1 \tag{38}$$

Substituting Eq. 2 and Eq. 38 into Eq. 8, the weight quantization loss after scaling can be written as:

$$E'_{\boldsymbol{W}_{:,1}} \approx \frac{1}{2}\mathbb{E}\left[\Delta W'_{1,1}{}^2 x'_1{}^2 + \Delta W'_{2,1}{}^2 x'_2{}^2\right]$$

$$\approx \frac{1}{2}\mathbb{E}\left[(\Delta W_{1,1} \cdot s_1)^2 (x_1/s_1)^2 + (\Delta W_{2,1} \cdot s_1)^2 (x_2/s_2)^2\right] \tag{39}$$

$$\approx \frac{1}{2}\mathbb{E}\left[\Delta W_{1,1}{}^2 x_1{}^2 + \frac{s_1{}^2}{s_2{}^2}\Delta W_{2,1}{}^2 x_2{}^2\right]$$

## B EXPERIMENT DETAILS

In this section, we present detailed experimental implementations. The pre-training models of VAR, RAR, PAR, and MAR are obtained from the official websites. PTQ4ARVG focuses on quantizing the decoder network. We empoly channel-wise asymmetric quantization for weights and layer-wise asymmetric quantization for activations. After scaling, PTQ4ARVG fuses all scaling factors into the network weights, ensuring zero additional overhead during inference. More specifically, we fuse the scaling factors into the AdaLN weights for RAR and VAR models, and into the weights of attention_norm and ffn_norm for the PAR and MAR models. For experimental evaluation, we use the ADM's TensorFlow evaluation suite *guided-diffusion* to evaluate FID, sFID, IS, and Precision.

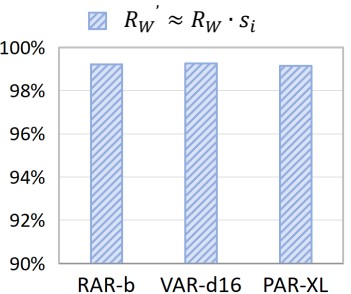

Figure 8: When $s_i > s_j$, over 99% of output channels satisfy $\big|R_W{}' - R_W \cdot s_i\big| < \big|R_W \cdot s_i \cdot 5\%\big|$.

## C BASELINE DETAILS

This section provides detailed implementation specifics for all baseline methods to facilitate reproducibility and comparison. The primary configuration settings are reported in Table 7. For scaling-based baselines, we apply them to all ARVG layers that can absorb scaling factors, specifically the $qkv$ and $fc1$ layers, and absorb the shifting and scaling factors into the network prior

to inference. For rotation-based methods, since ARVG cannot absorb rotation factors offline, the rotations are computed online during inference. All baselines, consistent with our method, quantize all linear layers and matrix multiplications, including the KV cache, and without relying on custom CUDA kernels. The baselines employ a randomly sampled calibration set of size 128. The specific implementation details are as follows:

**SmoothQuant.** SmoothQuant is a training-free method. In our reproduction, we strictly align all settings with the open-source implementation: channel-wise quantization for weights, layer-wise quantization for activations, and the default smoothing factor set to 0.5.

**OS+.** OS+ is a training-free method. We reproduce the standard OS+, applying channel-wise quantization for weights and layer-wise quantization for activations, without fine-tuning integration. All settings are strictly aligned with the open-source implementation: the shifting factors are computed from the per-channel min and max values; a grid search is used to determine the scaling threshold $t$ that minimizes output error, and channels exceeding $t$ are scaled to obtain the final scaling factors.

**RepQ.** RepQ is a training-free method. We reproduce RepQ based on the open-source implementation. The procedure is as follows: first, channel-wise activation quantization parameters are calibrated using the calibration set; next, shifting and scaling factors are computed to unify the activation ranges across channels; finally, the channel-wise activation quantization parameters are reparameterized into layer-wise parameters while preserving channel-wise weight quantization. For ARVG softmax activations, we use its $log\sqrt{2}$ quantizer.

**OmniQuant.** OmniQuant is a training-based method. Its trainable parameters include two weight clipping factors and two equivalent transformation factors. We retain the default parameter initialization and training architecture. The learning rates for the weight clipping factors and equivalent transformation factors are set to $5 \times 10^{-3}$ and $1 \times 10^{-2}$, respectively. Training is conducted with a batch size of 32 over 20 epochs on a calibration set of 128 images. We apply channel-wise quantization for weights and dynamic token-wise quantization for activations.

**SVDQuant.** SVDQuant is a training-free method. It first transfers activation outliers to the weights using a smoothing factor, then represents these outliers in a rank-$r$ branch matrix via singular value decomposition. For evaluation convenience, we do not integrate the branch matrix into the Nunchaku engine; instead, it is computed online during inference and added to the quantized weights. This simplification does not affect method performance. We set the smoothing factor to 0.5 and the rank to 32, using dynamic token-wise quantization for activations and channel-wise quantization for weights to match the default settings. Notably, while SVDQuant leaves some layers of diffusion models unquantized, in our reproduction, all ARVG layers are quantized to maintain consistency across all baselines and our method.

**QuaRot.** QuaRot is a training-free method. It mitigates activation outliers using rotation factors based on randomized Hadamard transforms. It exploits the rotation invariance of LayerNorm by applying a pair of inverse rotations to the layer's input and output, thereby reducing input outliers without altering the layer output. However, in ARVG, the adjacent layers use adaptive LayerNorm, which does not preserve rotation invariance. As a result, the rotation factors cannot be absorbed offline before inference. Therefore, we apply the rotation factors online during inference to handle outliers. Activations are quantized using dynamic token-wise quantization and weights using channel-wise quantization, consistent with the original method.

Table 7: Implementation details of baselines and PTQ4ARVG.

| Method | Weight Quant | Activation Quant | Training | Calibration | Online Compute |
|---|---|---|---|---|---|
| SmoothQuant | channel-wise | layer-wise | No | 128 random | No |
| OS+ | channel-wise | layer-wise | No | 128 random | No |
| RepQ* | channel-wise | layer-wise | No | 128 random | No |
| OmniQuant | channel-wise | dynamic token-wise | Yes | 128 random | No |
| SVDQuant | channel-wise | dynamic token-wise | No | 128 random | Yes |
| QuaRot | channel-wise | dynamic token-wise | No | 128 random | Yes |
| PTQ4ARVG | channel-wise | STWQ | No | 128 DGC | No |

## D  OVERVIEW OF PTQ4ARVG

In this section, we provide an intuitive visualization of PTQ4ARVG in Fig. 9. Our method addresses the challenges across different layers. It is worth noting that the quantization parameters in STWQ are statically set rather than calibrated dynamically during inference.

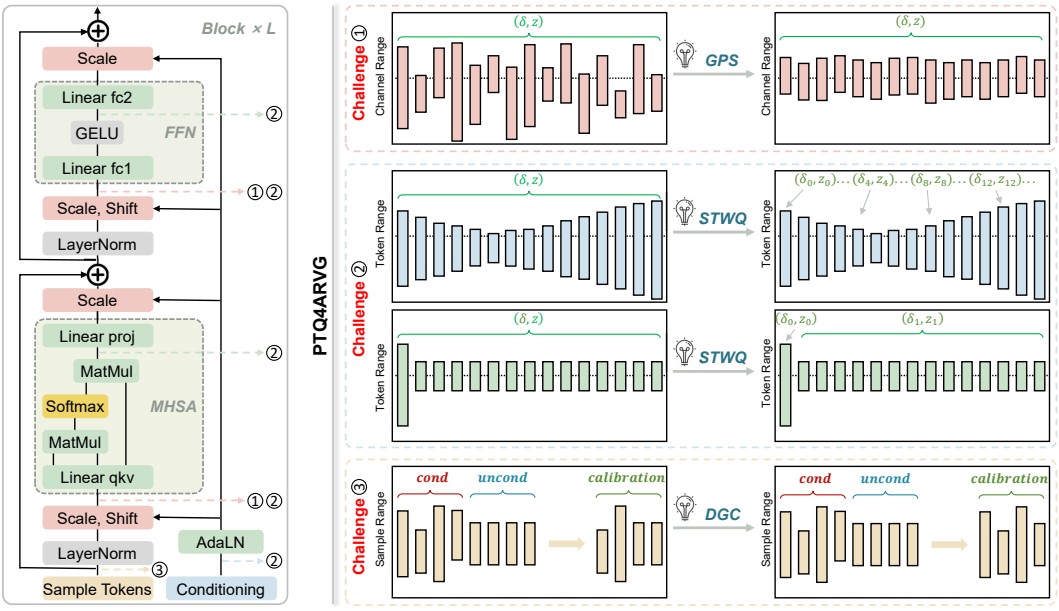

Figure 9: **Overview of PTQ4ARVG.**

## E  EMPIRICAL ANALYSIS OF APPROXIMATION BIASES

In this section, we conduct empirical analyses on the three approximations involved in GPS to demonstrate their validity and generality. The three approximations we analyze are:

(1) Hessian approximation of activation–weight quantization loss, omitting the MSE cross term. A theoretical justification for this approximation is provided in Appendix A.3.

(2) Upper-bound approximation of overall quantization error, where the activation quantization loss $\hat{E}_{\boldsymbol{x}} = \mathbb{E}\left[\mathcal{L}(\hat{\boldsymbol{x}}, \hat{\boldsymbol{W}}) - \mathcal{L}(\boldsymbol{x}, \hat{\boldsymbol{W}})\right]$ is approximated by $E_{\boldsymbol{x}} = \mathbb{E}\left[\mathcal{L}(\hat{\boldsymbol{x}}, \boldsymbol{W}) - \mathcal{L}(\boldsymbol{x}, \boldsymbol{W})\right]$. A theoretical justification for this approximation is provided in Appendix A.4.

(3) Scaling quantization error approximation, in which the estimated activation quantization error after scaling is used to approximate the true scaled quantization error. This approximation builds upon the error–estimation strategy validated in DilateQuant Liu et al. (2024c).

GPS is only applied to the $qkv$ and $fc1$ layers, where the scaling factors can be absorbed. We evaluate the approximation biases of these two layers across various models and input distributions. Specifically, we meansure this bias on RAR, VAR, PAR, and MAR using 64 samples. We compute the error (Approximations 2 and 3) using the L1 norm, while the loss (Approximation 1) is computed using the MSE to account for cross-term effects. As a result, the numerical values of the loss and the error approximations reported in the table differ. The results are summed over the sample and token dimensions and averaged over the channel dimension.

As shown in Table 8, all three approximations exhibit small biases across different models, layers, and input distributions, empirically supporting the validity of the assumptions underlying GPS.

## F  ALGORITHM OF GPS

The GPS can be implemented simply. Taking a linear layer as an example, GPS is illustrated in Algorithm 1. More importantly, since GPS inherently reduces quantization loss through stable scaling,

Table 8: Empirical analysis of approximation biases with 64 samples at INT6 setting. ($\cdot$%) denotes the ratio of the approximation bias to its ground-truth value. Here, the superscripts "real" and "appro" denote the measured true values and the approximations used in the paper, respectively. $E_{\boldsymbol{W}}^{bias}$ and $E_{\boldsymbol{x}}^{bias}$ represent the cross terms of weight-quant loss and activation-quant loss. "Bias" indicates the bias of the quantization error upper bound, and $\Delta \boldsymbol{x}'^{bias}$ denotes the difference between the true GPS-scaled activation quantization error and the estimated activation error according to our Eq. 10.

| Approximation | Object | RAR-B | | VAR-d16 | | PAR-XL | | MAR-B | |
|---|---|---|---|---|---|---|---|---|---|
| | | qkv | fc1 | qkv | fc1 | qkv | fc1 | qkv | fc1 |
| Hessian loss Approximation | $E_{\boldsymbol{W}}^{real}$ | 4.0080 | 8.6902 | 2.6107 | 10.3244 | 0.8325 | 6.6715 | 3.0049 | 63.1687 |
| | $E_{\boldsymbol{W}}^{appro}$ | 3.9422 | 8.6859 | 2.4609 | 10.2159 | 0.8183 | 6.4346 | 2.9558 | 59.9057 |
| | $E_{\boldsymbol{W}}^{bias}$ | 0.0659 | 0.0043 | 0.1498 | 0.1085 | 0.0143 | 0.2369 | 0.0491 | 3.2630 |
| | | (0.09%) | (0.05%) | (5.74%) | (1.05%) | (1.71%) | (3.55%) | (1.63%) | (5.17%) |
| | $E_{\boldsymbol{x}}^{real}$ | 6.8648 | 17.4328 | 6.8574 | 13.3940 | 1.5024 | 12.2295 | 7.3307 | 196.6933 |
| | $E_{\boldsymbol{x}}^{appro}$ | 6.8587 | 17.4188 | 6.6576 | 13.2114 | 1.4820 | 12.2041 | 7.3214 | 182.0671 |
| | $E_{\boldsymbol{x}}^{bias}$ | 0.0061 | 0.0140 | 0.1998 | 0.1826 | 0.0204 | 0.0254 | 0.0093 | 14.6262 |
| | | (0.09%) | (0.08%) | (2.91%) | (1.36%) | (1.36%) | (0.21%) | (0.13%) | (7.44%) |
| Upper-bound error Approximation | $\hat{E}_{\boldsymbol{x}}$ | 364.62 | 523.71 | 526.63 | 758.46 | 227.50 | 639.36 | 401.05 | 2077.32 |
| | $E_{\boldsymbol{x}}$ | 364.70 | 524.10 | 527.67 | 758.72 | 227.54 | 639.53 | 401.18 | 2077.61 |
| | $Bias$ | 0.0815 | 0.3931 | 1.0416 | 0.2679 | 0.0392 | 0.1740 | 0.1362 | 0.2905 |
| | | (0.02%) | (0.08%) | (0.20%) | (0.04%) | (0.02%) | (0.03%) | (0.03%) | (0.01%) |
| Scaling quantization error Approximation | $\Delta \boldsymbol{x}'^{real}$ | 92.23 | 108.80 | 158.57 | 137.33 | 183.05 | 245.78 | 145.28 | 440.34 |
| | $\Delta \boldsymbol{x}'^{appro}$ | 88.42 | 107.53 | 182.54 | 145.08 | 165.47 | 279.56 | 144.97 | 472.78 |
| | $\Delta \boldsymbol{x}'^{bias}$ | 3.8021 | 1.2688 | 23.9728 | 7.7560 | 17.5718 | 33.7800 | 0.3125 | 32.4476 |
| | | (4.12%) | (1.17%) | (15.12%) | (5.65%) | (9.60%) | (13.74%) | (0.22%) | (7.37%) |

---

**Algorithm 1** : Overall workflow of GPS

---

**Input**: activation $\boldsymbol{X} \in \mathbb{R}^{1 \times n}$ and $\boldsymbol{W} \in \mathbb{R}^{n \times m}$
**Output**: optimal scaling factor $\boldsymbol{s} \in \mathbb{R}^{n}$
**1. Preparing data**
$\quad \boldsymbol{X_q} = Q(\boldsymbol{X}), \boldsymbol{W_q} = Q(\boldsymbol{W})$ $\qquad\qquad\qquad\qquad$ ▷ obtain quantized values
$\quad \Delta \boldsymbol{X} = \boldsymbol{X} - \boldsymbol{X_q}, \Delta \boldsymbol{W} = \boldsymbol{W} - \boldsymbol{W_q}$ $\qquad\qquad$ ▷ calculate quantization errors
**2. Searching activation channel with the largest range**
$\quad \boldsymbol{R_x} = \max(\boldsymbol{X}, \dim = 0)[0] - \min(\boldsymbol{X}, \dim = 0)[0]$ $\qquad$ ▷ calculate activation range
$\quad R_x^k, k = \max(\boldsymbol{R_x})$ $\qquad\qquad\qquad$ ▷ find $k_{th}$ channel with the largest activation range
**3. Calculating $s_k$**
$\quad s_k = \sqrt{R_x^k / R_w^k}$ $\qquad\qquad\qquad\qquad\qquad$ ▷ calculate $s_k$ that ensures $R_x^{k'} = R_W^{k}{}'$
**4. Calculating remaining scaling factors based on $s_k$**
$\quad$ **for** $i = 1$ to $n$ **do**
$\quad\quad$ **if** $i = k$:
$\quad\quad\quad s_i = s_k$
$\quad\quad$ **else**:
$\quad\quad\quad s_i = s_k \dfrac{\sqrt{\sum_{j=1}^{m} |\Delta W_{i,j} x_i|}}{\sqrt{\sum_{j=1}^{m} |W_{i,j} \Delta x_i|}}$ $\qquad\qquad$ ▷ solve the scaling factor $s_i$ based on Eq. 16
$\quad$ **end for**
$\quad$ **return** $\boldsymbol{s}$

---

it can theoretically serve as a plug-and-play tool to enhance quantization performance across diverse frameworks. In future work, we will explore its generalization and applicability to other models.

## G VISUALIZATION OF GPS QUANTIZATION ERROR

Prior methods relied on empirically designed scaling factors to mitigate quantization error. In contrast, GPS derives the scaling factor from theoretical principles, explicitly modeling the quantization-error reduction brought by scaling and obtaining a closed-form optimal solution via differentiation. To visualize the advantages of GPS, we evaluate the quantization errors in the first block's $qkv$ and $fc1$ layers across different models (RAR-B, VAR-d16, PAR-XL, MAR-B), comparing the errors of no scaling, after applying SmoothQuant Xiao et al. (2023), after applying GPS, and after applying GPS with cross terms. As shown in Fig 10, GPS achieves a further reduction in quantization error compared with empirically designed scaling methods SmoothQuant. And retaining cross terms has a negligible impact on original GPS.

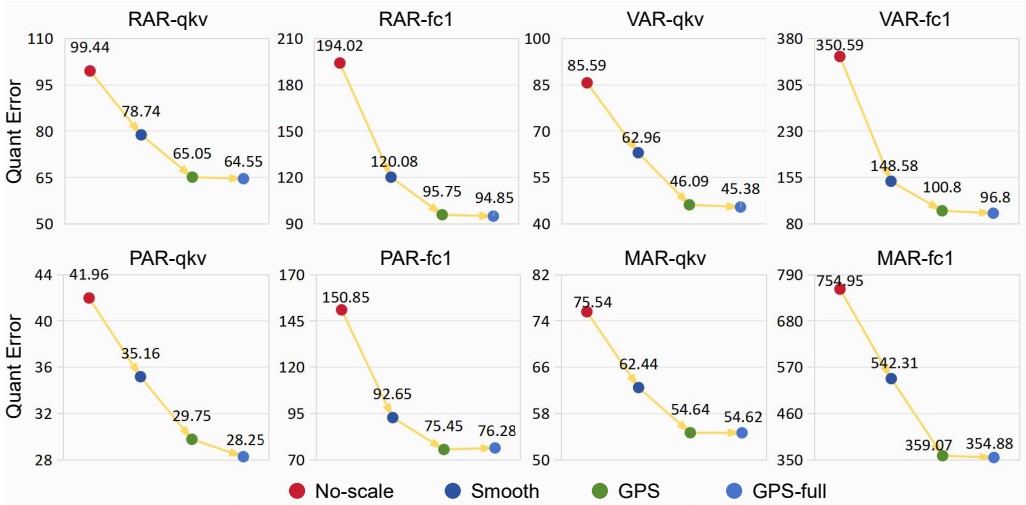

Figure 10: The quantization error of layer output under the INT6 setting using 64 samples. The error is measured with the L1 norm and averaged over tokens.

## H    ANALYSIS OF CHALLENGE 2

We analyze the underlying causes of highly dynamic activations at the token-wise level. Firstly, to ensure bidirectional dependencies among predicted image tokens, ARVG embeds *conditioning* information into the network via the AdaLN module. The *conditioning* includes not only the image conditional information but also the positional information of tokens. We observe that positional information varies across different position in the token sequence, resulting in **input of the AdaLN module showing variation along the token dimension**. Secondly, previous KV Cache studies on LLMs have identified that the initial token in Attention is highly sensitive to model performance, referring to these critical tokens as *sink* tokens. Unlike LLMs, we observe that in ARVG models, ***sink tokens are present in all linear layers of MHSA and FFN***. We attribute this property of ARVG to three key factors: (1) ARVG inherently uses class conditions as initial tokens, which encapsulate critical class information and play a pivotal role in conditional generation. (2) The initial tokens are visible to all subsequent tokens, making them readily trained to serve as highly sensitive tokens. (3) The distribution of the initial tokens are significantly different from that of all other tokens.

## I    VISUALIZATION OF POSITION-INVARIANT DISTRIBUTIONS IN TOKEN DIMENSION

To validate the position-invariant distributions of ARVG activations, we visualize token-wise activations for VAR-d16 and RAR-B across different layers, classes, and conditioning. Note that these models are trained on ImageNet and cannot generate images from other data distributions. Therefore, we do not verify this property on datasets beyond ImageNet. Since unconditional samples inherently correspond to class label 1000, we can merge the class and conditioning dimensions. For visualization, we select samples with labels 0, 1, 2, 999, and 1000 (unconditional), and visualize the distributions across different layers whenever possible. As shown in Fig. 11 and Fig. 12, the activations of VAR-d16 and RAR-B exhibit position-invariant distributions, remaining consistent across different classes and conditions. This confirms the motivation and implementation of STWQ.

## J    ADDITIONAL RESULTS

Table 9 shows the performance of PTQ4ARVG on RAR-L, RAR-XXL, VAR-d20, and VAR-d30. Similar to the PAR models, RAR-XXL does not satisfy the requirements of QuaRot, and thus no QuaRot experiments are conducted on this model. For VAR models, while SVDQuant performs well on the FID metric, it exhibits poor performance on the IS metric, likely because low-rank decomposition of the weights undermines the model's ability to generate diverse samples.

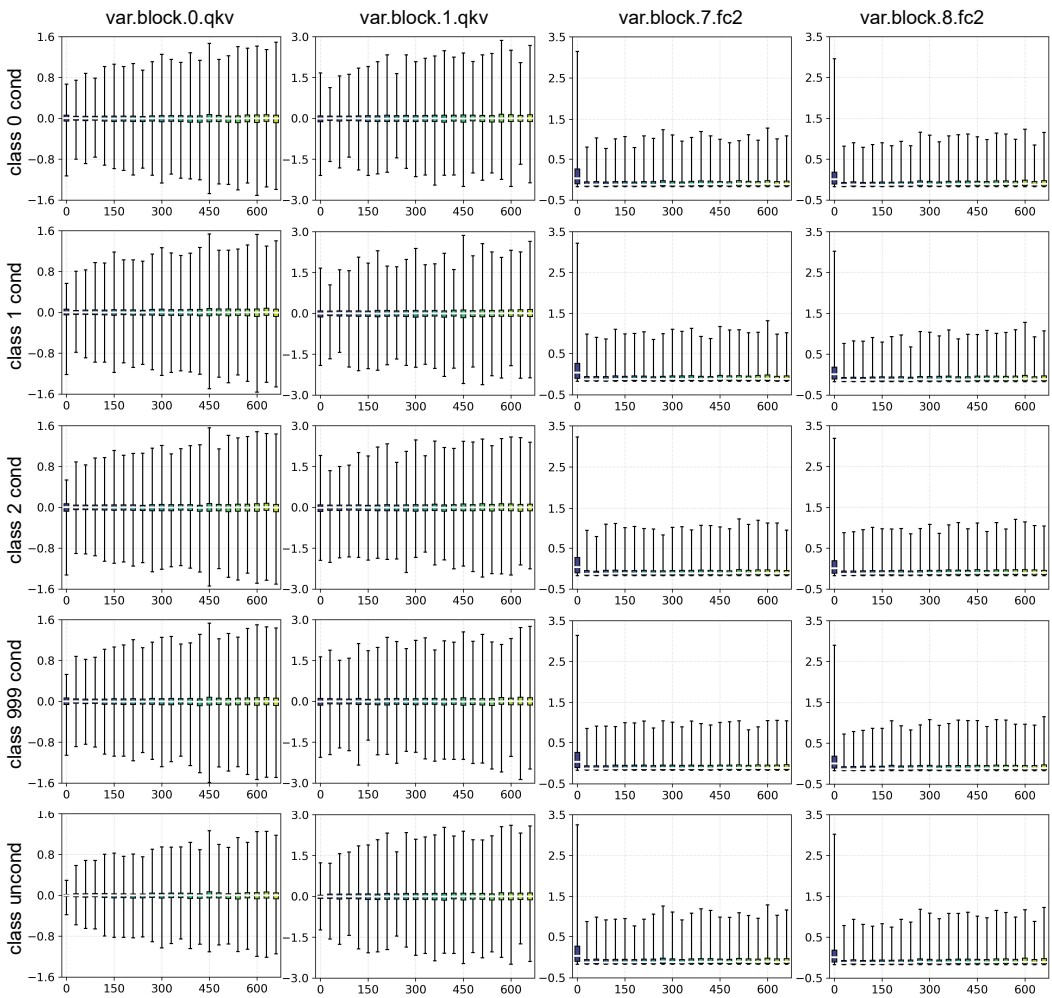

Figure 11: Visualization of token-wise activations in VAR-d16. Data come from a single sample.

Table 10 shows the performance of PTQ4ARVG on PAR-3B-4× and PAR-3B-16×.

While our approach demonstrates clear superiority at 6-bit, its 8-bit performance is on par with OmniQuant and SVDQuant for some models. To emphasize the advantages of our method, we include additional W4A8 experiments, with Table 11 further illustrating its superiority.

## K    COMPARISON WITH QUAROT

In some LLMs, the operation between two adjacent blocks is of the form $XW$, where $X$ denotes the output of the previous block and $W$ denotes the weights of the next block. When the two blocks are connected only by a normal LayerNorm (e.g., RMSNorm or LN), the rotation matrix $Q$ can be used offline, because of:

$$\text{LayerNorm}(X) = \text{LayerNorm}(XQ^T)Q \tag{40}$$

However, in VAR and RAR, the blocks are not only connected by a normal LayerNorm, but also by a specialized LayerNorm, AdaLN. AdaLN transforms the $conditioning$ input into modulation factors (MHSA$_{scale1}$, MHSA$_{shift1}$, MHSA$_{scale2}$, FFN$_{scale1}$, FFN$_{shift1}$, FFN$_{scale2}$ $\in \mathbb{R}^{T \times n}$), which adjust the distribution of activations. As illustrated in Fig. 9, assuming the MHSA output is denoted as $X$ and the residual as $X_r$, the computation of FFN.fc1 with weight $W$ is formulated as:

$$\text{output} = (\text{LN}(X \cdot \text{MHSA}_{scale2} + X_r) \cdot \text{FFN}_{scale1} + \text{FFN}_{shift1}) \cdot W \tag{41}$$

please refer to the VAR and RAR code for detail. As shown, applying rotation matrices offline does not preserve computational equivalence. Therefore, for VAR and RAR, rotation should be applied

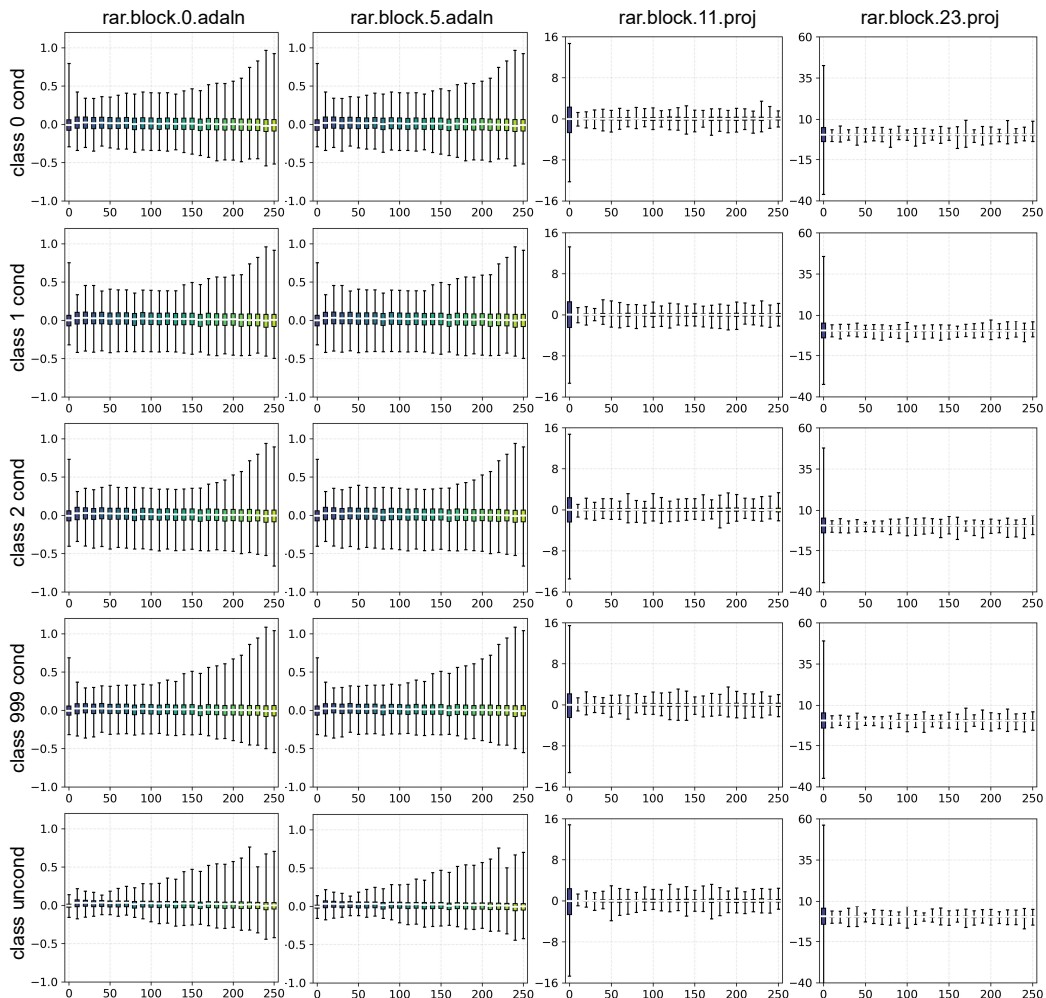

Figure 12: Visualization of token-wise activations in RAR-B. Data come from a single sample.

dynamically. The DiT model also employs AdaLN and shares a similar block structure with VAR and RAR. In ViDiT-Q, rotation matrices are also introduced to suppress outliers (although not explicitly mentioned in the paper). These rotation matrices are likewise applied dynamically, as shown in the ViDiT-Q code.

As a result, all Hadamard matrices involved in QuaRot need to be computed online when applied to RAR and VAR. This not only introduces significant overhead during inference but also increases peak memory usage. As reported in Table 12, with a batch size of 100 and a token sequence length of 256, QuaRot even leads to a 0.70× slowdown and a 159MB increase in peak memory usage at 8-bit quantization. Conversely, PTQ4ARVG achieves superior accuracy and quantization efficiency compared to QuaRot.

## L    LIMITATIONS

Recently, ARVG models demonstrates superior image generation capabilities compared to diffusion models. More importantly, their LLMs-compatible architecture and strong scaling laws make them a current focus of research. However, its deployment with the quantization techniques still remains largely unexplored. To address this gap, we propose PTQ4ARVG, an accurate and efficient post-training quantization framework tailored for the ARVG family.

Although our method can effectively quantize the weights and activations of ARVG models into 8-bit and 6-bit while preserving competitive performance, it struggles to maintain such a high level of

Table 9: Comparative results for RAR and VAR models.

| Bit Width | Methods | RAR-L | | | | VAR-d20 | | | |
|---|---|---|---|---|---|---|---|---|---|
| | | IS ↑ | FID ↓ | sFID ↓ | Precision ↑ | IS ↑ | FID ↓ | sFID ↓ | Precision ↑ |
| FP | - | 303.00 | 1.76 | 6.03 | 0.81 | 309.09 | 2.85 | 7.66 | 0.83 |
| W8A8 | SmoothQuant | 245.99 | 2.73 | 7.29 | 0.76 | 260.02 | 3.50 | 12.51 | 0.81 |
| | RepQ* | 253.16 | 2.55 | 7.34 | 0.76 | 258.90 | 3.46 | 11.98 | 0.79 |
| | OS+ | 252.98 | 2.70 | 7.94 | 0.67 | 260.96 | 3.40 | 11.25 | 0.81 |
| | OmniQuant | 289.61 | 2.16 | 6.66 | 0.79 | 239.30 | 4.14 | 11.73 | 0.77 |
| | QuaRot | 271.37 | 2.35 | 7.42 | 0.79 | 260.52 | 4.02 | 11.20 | 0.81 |
| | SVDQuant | 284.59 | 1.95 | 6.38 | 0.80 | 252.17 | 3.80 | 11.87 | 0.81 |
| | Ours | 291.55 | 1.90 | 6.34 | 0.81 | 263.86 | 3.36 | 11.17 | 0.82 |
| W6A6 | SmoothQuant | 31.97 | 63.34 | 40.03 | 0.39 | 190.96 | 5.21 | 10.61 | 0.71 |
| | RepQ* | 23.32 | 76.95 | 47.04 | 0.32 | 146.00 | 7.96 | 11.84 | 0.68 |
| | OS+ | 20.09 | 88.65 | 38.70 | 0.25 | 185.95 | 5.00 | 11.93 | 0.73 |
| | OmniQuant | 130.84 | 17.80 | 17.39 | 0.61 | 113.66 | 16.78 | 14.18 | 0.59 |
| | QuaRot | 45.27 | 53.27 | 34.09 | 0.45 | 190.73 | 5.02 | 10.68 | 0.74 |
| | SVDQuant | 200.65 | 5.28 | 8.06 | 0.71 | 151.88 | 7.50 | 11.47 | 0.64 |
| | Ours | 219.40 | 3.99 | 8.14 | 0.75 | 194.85 | 4.82 | 10.47 | 0.75 |
| Bit Width | Methods | RAR-XXL | | | | VAR-d30 | | | |
| FP | - | 328.87 | 1.51 | 5.13 | 0.81 | 307.24 | 2.03 | 8.72 | 0.81 |
| W8A8 | SmoothQuant | 278.89 | 2.09 | 5.91 | 0.77 | 247.74 | 4.37 | 18.19 | 0.76 |
| | RepQ* | 233.35 | 3.25 | 6.45 | 0.77 | 262.34 | 3.51 | 16.05 | 0.79 |
| | OS+ | 245.23 | 2.67 | 6.23 | 0.77 | 269.86 | 3.30 | 15.89 | 0.81 |
| | OmniQuant | 288.10 | 2.35 | 6.43 | 0.77 | 268.92 | 3.35 | 15.34 | 0.81 |
| | SVDQuant | 276.52 | 2.14 | 5.46 | 0.75 | 268.55 | 3.34 | 14.87 | 0.80 |
| | Ours | 321.03 | 1.61 | 5.28 | 0.82 | 277.05 | 3.27 | 14.40 | 0.81 |
| W6A6 | SmoothQuant | 164.15 | 12.47 | 18.74 | 0.67 | 101.01 | 22.67 | 32.05 | 0.59 |
| | RepQ* | 83.01 | 28.77 | 23.63 | 0.56 | 113.37 | 17.02 | 25.18 | 0.62 |
| | OS+ | 66.82 | 34.80 | 16.29 | 0.47 | 156.21 | 11.15 | 24.45 | 0.68 |
| | OmniQuant | 184.25 | 10.89 | 11.68 | 0.66 | 131.52 | 14.95 | 22.58 | 0.65 |
| | SVDQuant | 188.43 | 7.06 | 7.84 | 0.64 | 160.53 | 10.85 | 24.04 | 0.70 |
| | Ours | 266.39 | 2.41 | 5.70 | 0.77 | 168.84 | 8.50 | 21.38 | 0.71 |

Table 10: Comparative results for PAR models.

| Bit Width | Methods | PAR-3B-4× | | | | PAR-3B-16× | | | |
|---|---|---|---|---|---|---|---|---|---|
| | | IS ↑ | FID ↓ | sFID ↓ | Precision ↑ | IS ↑ | FID ↓ | sFID ↓ | Precision ↑ |
| FP | - | 255.5 | 2.29 | - | 0.82 | 262.5 | 2.88 | - | 0.82 |
| W8A8 | SmoothQuant | 10.91 | 112.24 | 40.94 | 0.10 | 12.00 | 79.45 | 78.58 | 0.16 |
| | RepQ* | 5.08 | 168.03 | 88.96 | 0.06 | 11.74 | 79.34 | 70.16 | 0.16 |
| | OS+ | 5.58 | 167.17 | 77.37 | 0.06 | 11.71 | 79.84 | 75.06 | 0.16 |
| | OmniQuant | 200.09 | 3.50 | 5.43 | 0.76 | 211.98 | 4.11 | 7.54 | 0.75 |
| | SVDQuant | 202.16 | 3.60 | 5.50 | 0.76 | 210.85 | 4.14 | 7.55 | 0.73 |
| | Ours | 200.42 | 3.57 | 5.73 | 0.76 | 207.48 | 4.16 | 6.86 | 0.73 |
| W6A6 | SmoothQuant | 9.87 | 137.03 | 42.37 | 0.06 | 9.49 | 107.07 | 96.20 | 0.11 |
| | RepQ* | 5.03 | 166.43 | 105.89 | 0.10 | 11.24 | 89.81 | 69.18 | 0.11 |
| | OS+ | 6.68 | 157.67 | 73.02 | 0.06 | 9.37 | 109.55 | 102.72 | 0.11 |
| | OmniQuant | 16.58 | 113.29 | 60.21 | 0.20 | 17.89 | 102.87 | 75.35 | 0.21 |
| | SVDQuant | 54.59 | 41.26 | 29.97 | 0.49 | 28.31 | 71.22 | 61.22 | 0.29 |
| | SQ+STWQ | 96.69 | 15.55 | 7.06 | 0.58 | 71.39 | 23.41 | 12.08 | 0.44 |
| | RepQ*+STWQ | 51.15 | 31.83 | 15.09 | 0.44 | 92.32 | 18.65 | 11.30 | 0.55 |
| | Ours | 121.91 | 10.05 | 6.22 | 0.63 | 107.52 | 15.47 | 10.72 | 0.58 |

accuracy when quantizing the model to 4-bit. Despite the limitations, we hope that our work could inspire the research interest on ARVG quantization within the community. We also further enhance our method to achieve better performance under lower bit-precision.

## M  ETHICS STATEMENT

This work proposes quantization-based acceleration methods for autoregressive visual generation models. It relies solely on publicly available datasets and does not involve human subjects, private data, or personally identifiable information. The methods are designed to improve computational efficiency and reduce energy consumption, thereby lowering the environmental cost of model deployment. We believe the ethical risks of this research are minimal.

Table 11: Comparative results at W4A8 precision.

| Bit Width | Model | Method | IS ↑ | FID ↓ | sFID ↓ | Precision ↑ |
|---|---|---|---|---|---|---|
| W4A8 | VAR-d16 | OmniQuant | 20.28 | 58.84 | 27.69 | 0.33 |
| | | SVDQuant | 88.34 | 19.26 | 22.50 | 0.57 |
| | | Ours | 114.09 | 17.06 | 16.99 | 0.58 |
| | RAR-B | OmniQuant | 125.77 | 15.86 | 16.13 | 0.59 |
| | | SVDQuant | 11.92 | 134.65 | 43.20 | 0.15 |
| | | Ours | 158.09 | 10.75 | 14.95 | 0.63 |
| | PAR-XL | OmniQuant | 7.40 | 133.55 | 43.76 | 0.13 |
| | | SVDQuant | 95.23 | 19.81 | 18.19 | 0.52 |
| | | Ours | 110.97 | 15.49 | 9.36 | 0.55 |

Table 12: Comparison with QuaRot on RAR-L with 8-bit quantization.

| Method | IS↑ | FID↓ | Precision↑ | Time (ms) | Memory (MB) | Speedup |
|---|---|---|---|---|---|---|
| FP | 303.00 | 1.76 | 0.81 | 3722 | 4241 | 1.00× |
| QuaRot | 271.37 | 2.35 | 0.79 | 6062 | 2345 | 0.70× |
| PTQ4ARVG | 291.55 | 1.90 | 0.81 | 1297 | 2186 | 2.87× |

## N    REPRODUCIBILITY STATEMENT

We have made every effort to ensure the reproducibility of our work. The paper provides detailed descriptions of the proposed methods, experimental setups, and evaluation protocols. Additionally, we provide complete source code and instructions in the supplementary materials.

## O    RANDOM SAMPLES

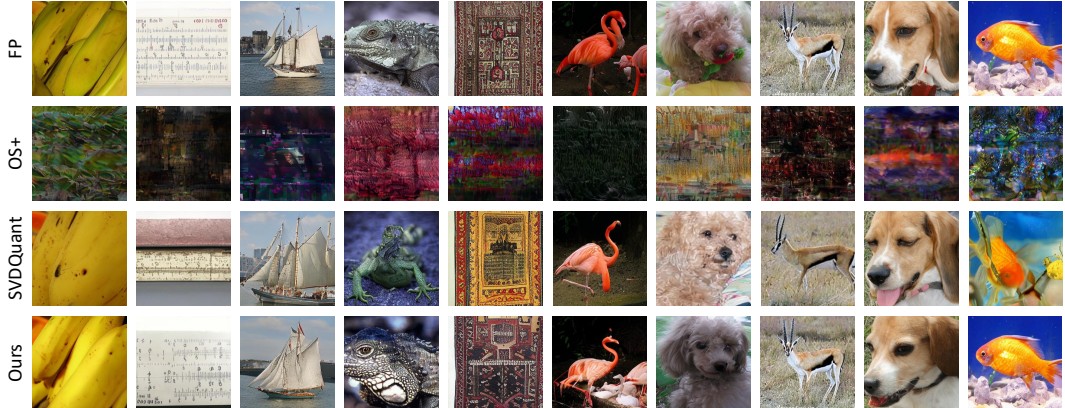

Figure 13: **Random samples of PAR-XL with 8-bit quantization.**

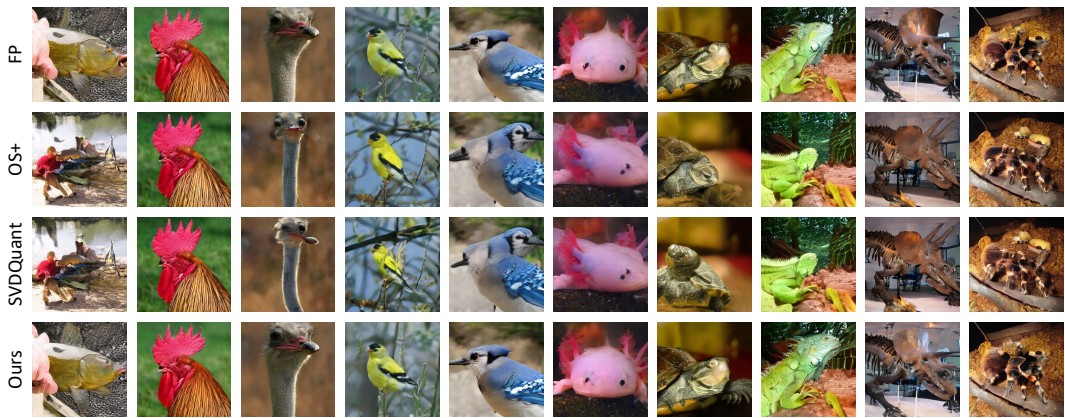

Figure 14: **Random samples of MAR-B with 8-bit quantization.**

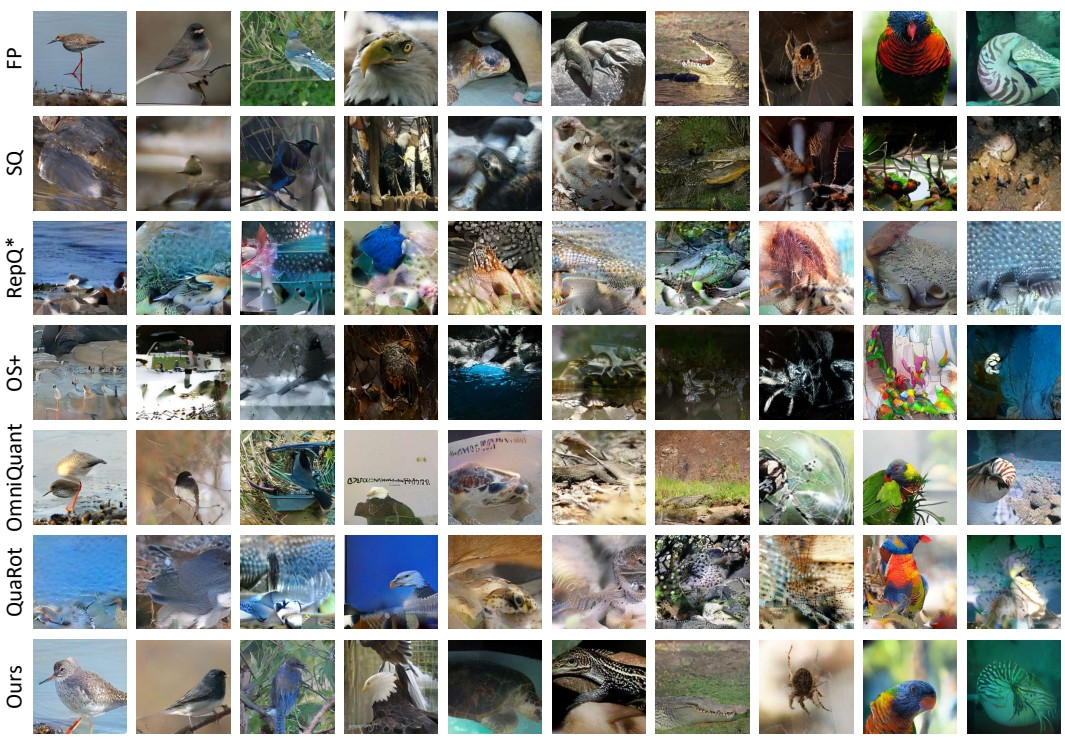

Figure 15: **Random samples of RAR-B with 6-bit quantization.**

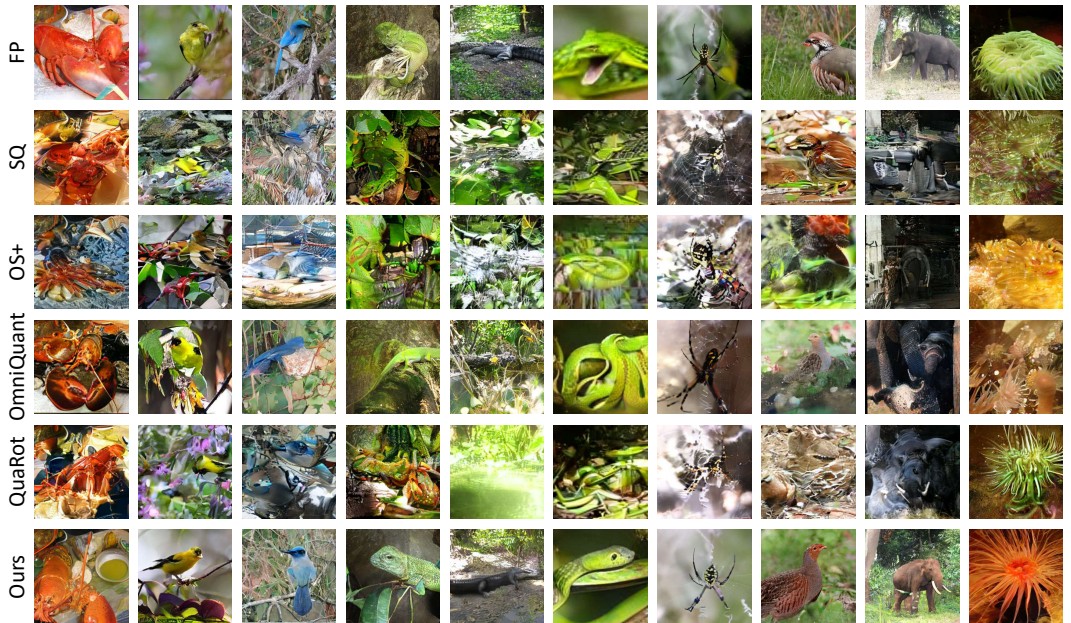

Figure 16: **Random samples of RAR-XL with 6-bit quantization.**

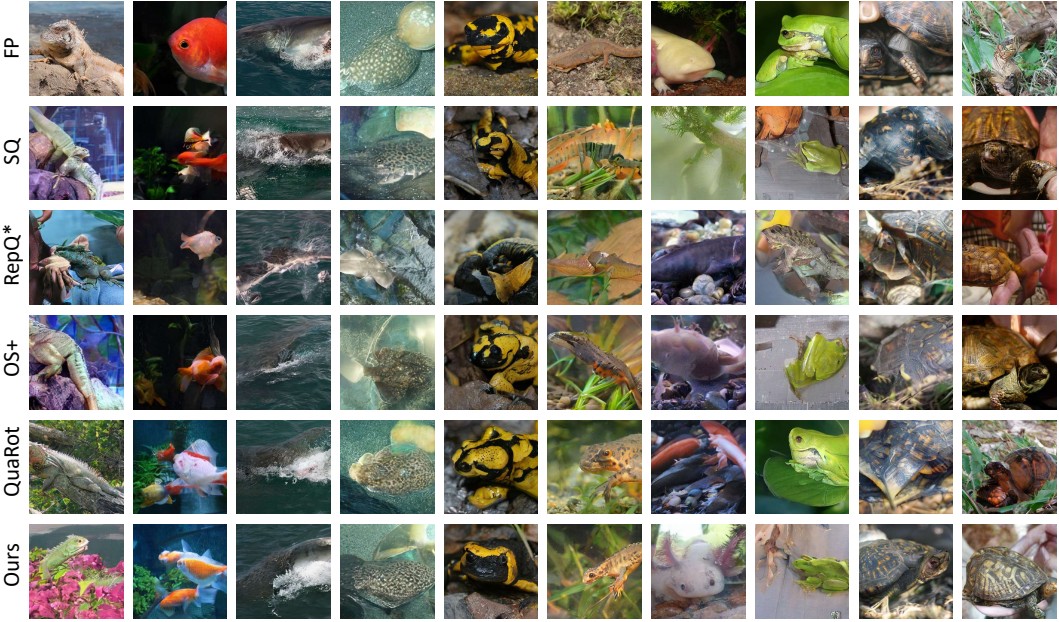

Figure 17: **Random samples of VAR-d16 with 6-bit quantization.**

