# OpenReview forum: "PTQ4ARVG: Post-Training Quantization for AutoRegressive Visual Generation Models"
_ICLR.cc/2026/Conference — ICLR 2026 Poster_

### Official Review · Reviewer_NhoA · 2025-10-24

**Soundness:** 2
**Presentation:** 2
**Contribution:** 3
**Rating:** 6
**Confidence:** 2

**Summary:**

The paper introduces PTQ4ARVG, the first post-training quantization framework for autoregressive visual generation (ARVG) models. It tackles channel outliers, token-wise dynamics, and sample-wise mismatches with Gain-Projected Scaling, Static Token-Wise Quantization, and Distribution-Guided Calibration, achieving strong 6/8-bit results and real GPU speedup over prior PTQ methods.

**Strengths:**

1. Novel problem setting, first systematic study of PTQ for ARVG models.
2. Clear motivation with three unique quantization challenges.
3. GPS provides a theory-driven scaling solution rather than heuristics.
4. STWQ and DGC are training-free and hardware-friendly.

**Weaknesses:**

1. Section 4.2 and 4.3 are relatively short and underdeveloped, lacking detailed analysis.
2. No visual comparison results are provided to qualitatively validate generation quality.
3. 8-bit results still show notable degradation, raising concerns about practical usability.
4. The reported 3.01× speedup at 8-bit is questionable without more deployment details.

**Questions:**

Please refer to the weakness section.

---

> ### Author Response · Authors · 2025-11-19
>
> Many thanks for the reviewer's valuable comments, which help us a lot to improve our work. We address your concerns as follows.
>
> ---
>
> > **W1:** Sec 4.2 and 4.3 are relatively short and underdeveloped, lacking detailed analysis.
>
> Thank you for the constructive comment.
>
> Due to space limitations, we condensed the descriptions in Sections 4.2 and 4.3. Accordingly:
>
> - We provide **additional analysis** related to Section 4.2 in **Appendix C**.
> - We include an **overall implementation workflow** covering both Sections 4.2 and 4.3 in **Appendix D**.
> - We conduct **independent ablation studies** for both components to validate their effectiveness.
>
> > **W2:** No visual comparison results are provided to qualitatively validate generation quality.
>
> We thank you for your valuable comment.
>
> Visual comparisons of different methods are presented in **Appendix K**, and we kindly refer you to these results.
>
> > **W3:** 8-bit results still show notable degradation, raising concerns about practical usability.
>
> Thank you for your suggestion regarding the practical applicability of our method.
>
> ARVG models exhibit higher sensitivity to quantization compared with LLMs and diffusion models, due to their **wider activation ranges** and **dynamic nature of token dimensions**.
>
> Specifically:
>
> * Compared with LLMs, whose activations in most layers remain within single-digit magnitudes, ARVG activations often reach **several tens or even hundreds**.
> * Compared with diffusion models, the autoregressive nature of ARVG induces highly **dynamic token-wise activations** across long sequences.
>
> Therefore, achieving nearly lossless INT8 quantization for ARVG is more difficult. In the INT8 setting, our method limits FID degradation to within **0.15** across all MAR models, and consistently below **0.25** on all RAR models.
> The visualizations in Appendix K further show that our method also preserves **strong semantic consistency** and **visual fidelity** on both PAR and VAR models.
>
> > **W4:** The reported speedup is questionable without more deployment details.
>
> We appreciate your careful and insightful comment.
>
> We provided deployment information, including the **execution platform** and **toolchain**, in the experimental settings and the appendix code.
> Specifically, compared with the standard deployment, we introduce **multiple quantization parameters** to support static token-wise quantization.

---

> > ### Comment · Reviewer_NhoA · 2025-11-25
> >
> > Thank you for the clarifications. I am satisfied with the responses and will maintain my score.

---

> > > ### Author Response · Authors · 2025-11-27
> > > **Response to Reviewer NhoA**
> > >
> > > Dear NhoA,
> > > Thank you very much for your prompt response. We are glad to hear that all your concerns have been addressed and sincerely appreciate your contribution to improving our paper. Please feel free to reach out if any further questions arise; we will remain actively engaged until the end of the rebuttal period.

---

### Official Review · Reviewer_FANZ · 2025-10-27

**Soundness:** 3
**Presentation:** 3
**Contribution:** 3
**Rating:** 4
**Confidence:** 4

**Summary:**

This paper is the first to systematically investigate the problem of post-training quantization (PTQ) for Autoregressive Visual Generation (ARVG) models. The authors identify three key challenges when directly applying existing PTQ methods to ARVG models: (1) severe channel-wise outliers, (2) highly dynamic token-wise activations, and (3) mismatched sample-wise distribution information. To address these challenges, the authors propose PTQ4ARVG, a training-free PTQ framework consisting of three core components: (1) Gain-Projected Scaling (GPS), which theoretically derives an optimal scaling factor to mitigate outliers by expanding the quantization loss via a Taylor series; (2) Static Token-Wise Quantization (STWQ), which leverages the fixed token length and position-invariant activation distribution of ARVG models to achieve fine-grained quantization without runtime overhead; and (3) Distribution-Guided Calibration (DGC), which selects more informative calibration samples by maximizing distributional entropy. Extensive experiments show that PTQ4ARVG can effectively quantize various ARVG models (e.g., VAR, RAR, PAR, MAR) to 8-bit and 6-bit, outperforming existing PTQ methods.

**Strengths:**

1. Pioneering and Important: This paper is the first comprehensive attempt to tackle the emerging and important problem of quantizing ARVG models, a largely unexplored area. With models like VAR and MAR gaining prominence, this work is very timely
2. Deep Problem Insight: The paper does not merely apply existing PTQ methods to a new model class. Instead, it deeply analyzes and identifies three specific, critical challenges (channel outliers, token dynamics, sample mismatch), strongly supporting these observations with visual evidence (Fig. 1). This in-depth analysis is a key strength
3. Solid Theoretical and Methodological Innovation: To address channel outliers, the proposed GPS method is not just empirical; it is mathematically modeled via a Taylor expansion of the quantization loss to derive a closed-form solution for the optimal scaling factor (Eq. 16). This is a solid and elegant theoretical contribution. Furthermore, STWQ is a clever solution that leverages the unique properties of ARVG models ("fixed token length" and "position-invariant distribution across samples") to solve the dynamic activation problem with a static, zero-overhead approach
4. Comprehensive Experimental Validation: The authors conduct extensive experiments on four different ARVG models (VAR, RAR, PAR, MAR) and compare against a wide range of state-of-the-art PTQ methods. The thorough ablation studies (Table 4, 5, 6) also clearly demonstrate the effectiveness of each proposed component

**Weaknesses:**

1. Performance Not Yet Optimal, Some 8-bit Results Are Not Near-Lossless: Although the proposed method outperforms other PTQ baselines, there is still room for performance improvement. For many quantization applications, 8-bit (W8A8) PTQ is often expected to achieve near-lossless performance compared to the full-precision (FP) model. However, the experimental results show that several models still exhibit a noticeable performance drop even at 8-bit. For instance, on VAR-d24, the FID degrades from 2.33 to 3.36, and on PAR-XL, it degrades from 2.61 to 3.55 (Table 1, Table 2). This suggests that ARVG models may be inherently more sensitive to quantization than other architectures, and achieving truly lossless performance remains a challenge even at 8-bit. This issue is, as expected, exacerbated at the more aggressive 6-bit setting, where the performance drop becomes much more significant (e.g., FID on RAR-B drops from 1.96 to 5.13).
 2. Absence of Comparison with QAT: The paper focuses exclusively on PTQ. While PTQ is valuable for its efficiency and training-free nature, providing a comparison with Quantization-Aware Training (QAT) would offer a more complete picture of the performance-cost trade-off. Even a simple QAT baseline (or even PEFT like Qlora) would help contextualize the performance level achieved by PTQ4ARVG.

**Questions:**

1. Regarding the GPS Assumption: In the derivation of GPS, you assume that the Hessian of the loss function with respect to the layer's output can be treated as a common constant and thus ignored during optimization. This is a strong simplification. Could you please elaborate on the validity of this assumption? Have you analyzed the actual variation of this Hessian value in ARVG models? How sensitive is the performance of GPS to this assumption?
2. Pareto Front Analysis vs. Smaller Full-Precision Models: As this is a pioneering work in PTQ for ARVG models, it is crucial to establish the practical benefits of quantization against the primary alternative for efficiency: training smaller, full-precision models. Does a quantized large model (e.g., a 6-bit RAR-XL) offer a better trade-off than a smaller full-precision model (e.g., a FP RAR-B) with a similar FLOPs count? A comparison on a FLOPs-vs-FID plot would be highly valuable to demonstrate that the proposed quantization method truly pushes the Pareto front, rather than just landing on a point that could be achieved by a smaller, unquantized model.
3. On the Composability of the Methods: The proposed methods are designed to tackle specific challenges in ARVG. We are interested in understanding the composability of these methods with other advanced PTQ techniques, such as low-rank decomposition from SVDQuant or rotation-based methods from QuaRot. For instance, could combining GPS with these approaches lead to further performance gains, potentially achieving near-lossless results at 6-bit and 8-bit? This is crucial for assessing the generality and future potential of your framework
$\textbf{If these issues can be resolved, I will consider giving a higher score.}$

---

> ### Author Response · Authors · 2025-11-19
>
> Many thanks for the reviewer's valuable comments and questions, which help us a lot to improve our work. We address your concerns as follows.
>
> ---
>
> > **W1:** Performance Not Yet Optimal, Some 8-bit Results Are Not Near-Lossless.
>
> We sincerely appreciate your insightful and detailed comments.
>
> As you pointed out, ARVG models indeed exhibit higher sensitivity to quantization compared with LLMs and DMs.
> The underlying cause lies in their **wider activation ranges and dynamic token dimensions**:
>
> - Compared with LLMs, whose activations in most layers remain within single-digit magnitudes, ARVG activations often reach tens or even hundreds, resulting in **larger quantization mapping errors**.
> - Compared with DMs, the autoregressive nature of ARVG produces highly dynamic token-wise activations across long sequences, rendering **layer-wise static quantization parameters difficult to adapt effectively**.
> - Even with dynamic token-wise quantization, the extreme outliers and broader activation ranges of visual tokens lead **min–max calibration to produce significant errors**.
>
> Consequently, **ARVG is more difficult to quantize under INT8** compared to LLMs and DMs, causing state-of-the-art PTQ methods to suffer notable performance drops and, in certain cases, produce unusable results.
>
> To address this unique challenge, we introduce PTQ4ARVG, which significantly improves their **stability and usability** under INT8 quantization, evidenced by limiting FID degradation to within **0.15** across all MAR models and consistently below **0.25** for all RAR models.
> The visualizations in **Appendix K** further demonstrate that our INT8-quantized models consistently retain usable generation quality.
>
> > **W2:** Absence of Comparison with QAT.
>
> Thanks to your professional comment.
>
> Following your suggestion, we have compared our method with the advanced **QAT baseline EfficientDM** [1] and added a **performance–cost trade-off analysis**. The results demonstrate the advantages of our method in efficiency.
> We kindly invite you to review them in **Appendix R**.
>
> [1] EfficientDM: Efficient Quantization-Aware Fine-Tuning of Low-Bit Diffusion Models. ICLR 2024.

---

> ### Author Response · Authors · 2025-11-19
>
> > **Q1:** The impact of the Hessian loss function on GPS.
>
> We thank you for your detailed and valuable comments.
>
> We perform a Taylor expansion of the quantization loss to obtain a Hessian-based formulation, which we further approximate as an MSE loss based on our derivation. Here, we introduce the Hessian task loss $\bigtriangledown^2_{\mathbf{y_k}} \mathcal{L}$. Since the quantization loss is evaluated using MSE, $\bigtriangledown^2_{\mathbf{y_k}} \mathcal{L}$ is treated as a constant. For example, when the quantization loss is written as $\mathcal{L} = \tfrac{1}{2}(y_k - y_k^q)^2$, where $y_k^q$ denotes the quantized output and $y_k$ denotes the full-precision output, the second derivative of $\mathcal{L}$ with respect to $y_k$, i.e., $\bigtriangledown^2_{\mathbf{y_k}} \mathcal{L}$, is the constant 1.
> Given that PyTorch can only represent first-order derivatives and is constrained by floating-point precision, we emphasize in the paper that $\bigtriangledown^2_{\mathbf{y_k}} \mathcal{L}$ is unobservable and may exhibit slight numerical discrepancies. Thus, **it has negligible impact on the theoretical derivation and practical implementation of GPS**.
>
> > **Q2:** Computation-Performance trade-off between the quantized model and smaller full-precision models.
>
> Thank you for your constructive comment.
>
> Following your suggestion, we explore the trade-off between computation and performance for quantized models vs. retrained smaller models.
> We attempted to retrain RAR-B on 2 H100 GPUs, aligning with the original training setup of 1.28M images for 500 epochs, which would require 6 days. Due to these resource and time constraints, we were unfortunately unable to perform retraining on smaller models.
>
> However, to address your question, we explore this trade-off using existing models. Specifically, **we evaluate the computation and performance of models of different sizes before and after quantization, with particular focus on comparisons between smaller full-precision models and larger quantized models**. The results show that model quantization offers more practical
> efficiency benefits than smaller full-precision models. We have added this analysis to the paper. Please refer to **Appendix S** for details.
>
> > **Q3:** On the Composability of the Methods.
>
> We appreciate your insightful comment.
>
> Following your suggestion, we combine GPS with the low-rank decomposition method SVDQuant to further validate the generality of our approach and its potential for improving performance.
>
> As shown in the Table, our method **improves the performance of SVDQuant, further enabling nearly lossless results at 6-bit and 8-bit settings**.
>
> | Model| Bit Width | Method | IS ↑ | FID ↓| sFID ↓  |
> |------------|-----------|----------------|----------|---------|---------|
> |  VAR-d16  | W8A8      | SVDQuant       | 229.36    | 4.11    | 12.72   |
> | | W8A8      | SVDQuant+GPS   | **284.52**    | **3.62**    | **10.35**   |
> | | W6A6      | SVDQuant       | 130.15    | 12.53   | 15.36   |
> | | W6A6      | SVDQuant+GPS   | **150.26**    | **9.97**    | **13.45**   |
> |  RAR-L  | W8A8      | SVDQuant       | 284.59   | 1.95    | 6.38    |
> | | W8A8      | SVDQuant+GPS   | **285.47**   | **1.92**    | **6.22**    |
> | | W6A6      | SVDQuant       | 200.65   | 5.28    | 8.06    |
> |  | W6A6      | SVDQuant+GPS   | **210.53**   | **4.22**    | **7.87**    |
> | PAR-XL-4x| W8A8      | SVDQuant       | 213.98   | 3.80    | 7.79    |
> | | W8A8      | SVDQuant+GPS   | **224.84**   | **3.10**    | **6.74**    |
> | | W6A6      | SVDQuant       | 102.03   | 19.52   | 18.19   |
> |   | W6A6      | SVDQuant+GPS   | **114.80**   | **12.45**   | **7.25**    |

---

> > ### Comment · Reviewer_FANZ · 2025-11-25
> >
> > Thank your response. Below are the key issues that still need to be addressed.
> >
> > 1. SVDQuant Baseline Performance & Implementation. SVDQuant baseline performance is suspiciously low, potentially due to an implementation error (using Per-channel instead of Per-group 64 quantization).
> > $\textbf{Suggestion}$: Please provide Per-group 64 quantization results or the implementation code to confirm baseline accuracy.
> >
> > 2. Method Composability & "Lossless" Definition. Issue: Current composability results still show significant loss at 8-bit/6-bit, making the "nearly lossless" claim confusing.The existing combined scheme, even with SVDQuant, fails to achieve lossless performance at 8-bit/6-bit, which is often unacceptable.
> >  $\textbf{Suggestion}$: Please provide a combined scheme that truly achieves lossless performance at 8-bit/6-bit in order to demonstrate the upper limit of your method.
> >
> > 3. QAT Baseline Fairness. Issue: The fairness of the comparison against the QAT baseline (EfficientDM) is questioned, suspecting inconsistent activation quantization granularity (Layer-wise vs. Token-wise).
> > $\textbf{Suggestion}$: Please confirm the activation quantization configuration for EfficientDM(token-wise activation quantization) to ensure a fair comparison with your GPS method. You don't need to keep default setting for EfficientDM.
> >
> > I look forward to receiving further experiments and explanations addressing these points.

---

> ### Author Response · Authors · 2025-11-27
> **Further Response to Reviewer FANZ**
>
> Dear FANZ,
>
> Thank you for your active response. We greatly appreciate the opportunity to further address your concerns and are especially grateful for your detailed suggestions, which have significantly helped us improve our paper. We address your remaining concerns as follows:
>
> ---
>
> > 1: SVDQuant baseline performance is suspiciously low, potentially due to an implementation error (using Per-channel instead of Per-group 64 quantization).
>
> Thank you for your constructive suggestion. **SVDQuant adopts per-channel weight quantization (8-bit/6-bit) in its original paper. Accordingly, we retain this default quantization granularity in our ARVG reproducibility experiments**. Finer-grained per-group quantization can indeed improve quantization accuracy, particularly for weight quantization.
> However, our ablation studies reveal that the poor performance of many existing methods on ARVG primarily stems from activation quantization.
> Specifically, as shown in Table, using the SmoothQuant without quantizing activations preserves nearly lossless performance even when weights are quantized to 6-bit per channel.
> Based on this observations, applying per-group quantization to SVDQuant is also unlikely to change its performance.
>
>
> |Model|Bits|IS|FID|sFID|
> |----|----|----|---|---|
> |VAR-d16|W16A16|283.21|3.60|8.27|
> | |W6A16|280.37|3.39|8.49|
> |RAR-B|W16A16|292.80|1.96|6.16|
> | |W6A16|267.71|2.63|7.11|
> |PAR-XL|W16A16|259.20|2.61|6.30|
> | |W6A16|236.36|2.59|5.80|
>
> To address your concern, we re-examined the implementation of SVDQuant and identified two main reasons for its suboptimal performance on ARVG:
>
> - SVDQuant mitigates activation outliers in the `qkv` and `fc1` layers primarily through low-rank decomposition, while **retaining full precision for the quantization-sensitive proj layers in the original paper**. In our setup, we **quantize all layers** for all baseline methods.
> The stronger token-wise activation dynamics in ARVG’s `proj` and `fc2` layers, **where low-rank decomposition cannot be applied**, lead to the performance drop of SVDQuant.
> - SVDQuant is tailored to the activation distributions of diffusion models, **limiting its cross-architecture adaptability**. Existing quantization study FBQuant[1] shows that **SVDQuant underperforms autoregressive-based methods such as OmniQuant on multiple LLMs quantization tasks**. This indicates that SVDQuant cannot retain its advantages in ARVG.
>
> [1] Fbquant: Feedback quantization for large language models.
>
>
> > 2: Current composability results still show significant loss at 8-bit/6-bit, making the "nearly lossless" claim confusing.
>
> We sincerely thank you for your detailed and constructive suggestions. Regarding the term “nearly lossless”, we would like to clarify the following points:
>
> - **ARVG models face more severe quantization challenges** than diffusion models and LLMs. For instance, under 8-bit quantization, SmoothQuant maintains nearly lossless performance on diffusion models and LLMs, whereas its performance on ARVG drops significantly, sometimes even collapsing.
>
> - **We have evaluated state-of-the-art quantization methods on ARVG**, including the training-based OmniQuant, rotation-based QuaRot, and low-rank SVDQuant. Even so, these baseline methods still exhibit significant performance drops under 8-bit quantization, highlighting the intrinsic difficulty of quantizing ARVG.
>
> - **our method significantly outperforms all existing state-of-the-art baselines**, demonstrating its effectiveness and tailored design for ARVG quantization.
>
> - Regarding the realization of a “truly lossless” combined strategy, we believe it is currently difficult to achieve. Existing quantization schemes are not compatible with the activation distributions and autoregressive structure of ARVG, and no ARVG-specific quantization methods can be directly combined. Our method, **tailored to ARVG characteristics, performs targeted quantization comprehensively over channel, token, and sample dimensions**. Experimental results demonstrate that under INT8, **our method is nearly lossless: FID drops are <0.1 for MAR, <0.25 for RAR, and <1.0 for most PAR and VAR models.**
>
>
> > 3: QAT Baseline Fairness. You don't need to keep default setting for EfficientDM.
>
> In response to your valuable suggestion, we configured EfficientDM with per-token activation quantization and retrained it accordingly. We kindly invite you to review them in **Appendix R**.

---

### Official Review · Reviewer_x7xs · 2025-10-31

**Soundness:** 3
**Presentation:** 3
**Contribution:** 2
**Rating:** 6
**Confidence:** 2

**Summary:**

PTQ4ARVG addresses the underexplored challenge of applying Post-Training Quantization (PTQ) to Autoregressive Visual Generation (ARVG) models , which share architectural similarities with large language models but achieve visual performance comparable to diffusion-based models. The paper identifies three key obstacles specific to ARVG models that cause conventional PTQ methods to fail: (1) severe outliers at the channel-wise level (due to AdaLN-adjusted activations), (2) highly dynamic activations at the token-wise level (due to positional embedding and sink tokens), and (3) mismatched distribution information at the sample-wise level. To overcome these, the authors propose a tailored PTQ framework, PTQ4ARVG, which consists of Gain-Projected Scaling (GPS), Static Token-Wise Quantization (STWQ), and Distribution-Guided Calibration (DGC). The ultimate goal is to significantly reduce model size and inference latency, enabling efficient deployment of large ARVG models on resource-constrained devices without the need for extensive retraining.

**Strengths:**

1. High Relevance and Motivation: The work addresses a critical bottleneck—quantization—for deploying large generative models, filling a significant research gap within the emerging ARVG model class.

2. Clarity and Technical Detail: The paper clearly articulates the unique quantization challenges specific to ARVG and proposes technically sound solutions, such as GPS, which optimizes the scaling factor using a closed-form solution derived from a Taylor series expansion.

3. Extensive Robustness Validation: The method is evaluated across a broad family of state-of-the-art ARVG models (VAR, RAR, PAR, MAR), demonstrating consistent superiority over existing PTQ methods at both W8A8 and W6A6 bit-widths.

**Weaknesses:**

The generative evaluation is confined to the ImageNet dataset and relies heavily on traditional metrics like FID/IS which correlate poorly with human perception, suggesting that the inclusion of modern perceptual metrics (e.g., HPS [1] or CLIP Score) on diverse, high-fidelity datasets is strongly recommended for a more robust comparison.

[1] Ma Y, Wu X, Sun K, et al. Hpsv3: Towards wide-spectrum human preference score. CVPR 2025: 15086-15095.

**Questions:**

None

---

> ### Author Response · Authors · 2025-11-19
>
> Many thanks for the reviewer's valuable comments, which help us a lot to improve our work. We address your concerns as follows.
>
> ---
>
> > **W1:** The generative evaluation is confined to the ImageNet dataset and relies heavily on traditional metrics like FID/IS.
>
> We appreciate your constructive suggestions.
>
> Although current generative model metrics still have notable limitations, FID and IS remain the most widely adopted standards for fair comparison, as no universally accepted alternatives have yet emerged.
> **To ensure consistency with the original model reports, we follow the same evaluation**.
> Moreover, because the model is trained solely on ImageNet, its generative distribution does not align with other datasets, making cross-dataset evaluation less meaningful.
>
> Following your suggestion, we additionally report the advanced metric HPS. As shown in table, **the superiority of our method becomes more pronounced when assessed with more accurate evaluation metrics**. Here, FP refers to the full-precision model, and SVDQuant is the strong quantization baseline used for comparison with our method.
>
> | Method    | VAR-d16 HPS | RAR-B HPS | PAR-XL HPS |
> |-----------|------------|-----------|------------|
> | FP        | 2.48       | 0.74      | 3.29       |
> | SVDQuant  | 1.24       | 0.21      | 3.13       |
> | Ours      | 1.65       | 0.56      | 3.26       |

---

### Official Review · Reviewer_oG1L · 2025-11-01

**Soundness:** 3
**Presentation:** 3
**Contribution:** 3
**Rating:** 4
**Confidence:** 4

**Summary:**

This paper studies PTQ for ARVG models. The authors empirically identify three categories of quantization challenges for ARVG: severe channel-level outliers, highly dynamic activations along the token dimension, and sample-level distribution mismatch. To address these, they propose the PTQ4ARVG framework, which comprises three components, GPS, STWQ, and DGC. Experiments cover multiple ARVG models (VAR, RAR, PAR, MAR), and comparisons under W8A8 and W6A6 settings against baselines such as SmoothQuant, OS+, OmniQuant, QuaRot, and SVDQuant show advantages in generation quality and inference acceleration.

**Strengths:**

1. Problem formulation is clear and well targeted, The authors carefully analyze how ARVG differs from standard LLMs and diffusion models in activation distributions and token structure, The three proposed challenges at the channel, token, and sample levels capture the essential difficulties of quantizing ARVG, providing a solid basis for method design.
2. Method design has theoretical support, The GPS component is not purely heuristic, it derives an analytic expression for the effect of the scaling factor via decomposition of the quantization loss and a Taylor expansion, and it yields a closed form or solvable expression. This is more convincing than many scaling methods that rely only on heuristic statistics.
3. Practicality and deployment considerations are thorough, The STWQ staticization idea matches ARVG’s fixed sequence length property, avoiding dynamic online calibration overhead, and the authors demonstrate deployment with standard CUDA kernels and real latency and memory measurements, which strengthens the engineering credibility of the work.
4. Comprehensive experimental coverage and ablation, The paper compares several mainstream ARVG models across multiple bit widths (8/6 bit), and provides component-wise ablations for GPS, STWQ, and DGC, showing each component’s contribution to overall performance, The empirical comparison is relatively systematic.

**Weaknesses:**

1. Several important approximations and assumptions in GPS are not sufficiently validated, GPS omits Hessian cross terms in its derivation, however prior series of quantization works (OBD[1], OBS[2], OBC[3], GPTQ[4]) have pointed out that such omissions can introduce significant errors.
2. The position-invariance assumption underlying STWQ is limited, The authors rely on “position-invariant distributions” as the core justification for STWQ, but the paper only shows statistics for a few layers and a few sample types (Fig.4), There is insufficient validation across models, classes, or conditioning states, If some classes or conditioning strongly affect position distributions, STWQ’s effectiveness may degrade.
3. Baseline comparisons and fairness of hyperparameter / implementation choices are unclear, Several baselines compared (notably OmniQuant, QuaRot, SVDQuant) have multiple implementations and hyperparameter variants in the literature, The paper states “default settings” but does not specify reproduction details, calibration sample counts, or whether fine-tuning was permitted, Some baselines show anomalously poor performance in the tables, which could indicate inconsistent setups or implementation issues, this undermines confidence in the conclusions.
4. Some result presentations and tables lack readability and consistency, Tables 1/2/3 contain many anomalous or extreme values (for example SVDQuant’s IS collapse under W8A8 in Table 1), but the paper does not explain or annotate these anomalies, Table headers and notes are insufficiently detailed, making it hard to trace exact experimental settings such as batch size, calibration sample count, sequence length, and whether embedding / KV-cache were quantized.
5. The method already fails to preserve accuracy compared to full precision at 8 bits in most settings, this greatly affects practicality, The paper also does not explore more aggressive quantization at 4 bits or below.

References:

1. LeCun et al., Optimal Brain Damage.
2. Hassibi et al., Optimal Brain Surgeon and general network pruning.
3. Elias et al., Optimal Brain Compression.
4. Frantar et al., GPTQ: Accurate Post-Training Quantization for Generative Pre-trained Transformers.

**Questions:**

1. How large is the effect of omitting Hessian cross terms and assuming Hessian constancy in GPS on the final scaling factors? Can you provide real Hessian statistics or approximations (diagonal, top-k) for several representative layers, such as MHSA, FFN, and projection layers, to validate the second-order Taylor approximation and the omission of cross terms? If the assumptions are invalid, by how much does GPS’s closed-form solution deviate?
2. Does STWQ’s "position-invariant distribution" hold across different models, different conditioning modes (conditional vs unconditional), and different datasets (ImageNet vs more complex scenes)? I suggest providing cross-model and cross-dataset statistics, or at least verifying position distribution stability on the classification/conditioning subsets of PAR, MAR, and VAR.
3. Please provide full reproduction details for baseline implementations and comparisons, including the specific implementation versions, calibration sample counts used for each method, whether KV-cache / embeddings / layernorm parameters were quantized, and whether custom CUDA kernels were used. If possible, include key reproduction scripts or hyperparameter tables in an appendix.
4. For example, in Table 1 the VAR-d16 W8A8 IS has already dropped by nearly 20%, while historically 8-bit quantization is often considered effectively lossless, Why then does the paper still claim maintained competitive performance in the abstract? Please clarify and reconcile these claims with the observed degradations.
5. GPS numerical stability, Formula (16) includes square roots and denominators, In channels with sparsity or where Δx approaches zero, might this lead to numerical instability? What regularization or clipping strategies are used to prevent division by near-zero values?
6. STWQ vs DTWQ trade-off, STWQ avoids online overhead but how does it handle variable sequence lengths or deployment scenarios requiring arbitrary cropping? Do you need to rebuild static tables for different sequence lengths, or is there an adaptive strategy?
7. DGC’s Mahalanobis thresholding, the paper fixes “top 50%” of samples as calibration picks, how was this threshold chosen? Does it need tuning across models or datasets, and is there a more automatic threshold selection method, for example based on distributional entropy elbow points?

---

> ### Author Response · Authors · 2025-11-19
>
> Many thanks for the reviewer's valuable comments and questions, which help us a lot to improve our work. We address your concerns as follows.
>
> ---
>
> > **W1 & Q1:** Several important approximations and assumptions in GPS are not sufficiently validated. How large is the effect of omitting cross terms in GPS.
>
> Thank you for your valuable comment.
>
> GPS relies on a assumption (Remark 1). As detailed in Sec. 4.1 of the paper, we have **validated this assumption via comprehensive statistical analyses across multiple ARVG models**.
>
> GPS also involves three approximations, for which we **provide theoretical justifications regarding their validity and accuracy in Appendices A.3 and A.4**.
>
> To address your concern, we have included **comprehensive statistical analyses of the bias associated with each approximation in Appendix L**. We kindly invite you to review them.
>
> The results indicate that **the omitted cross terms contribute only a negligible fraction of the total quantization loss**. For instance, in the `qkv` and `fc1` layers of RAR-B, their contribution is below **0.1%**. Given this minimal impact, we consider their effect on GPS to be negligible.
>
> Moreover, OBD explicitly eliminates full cross terms, whereas OBS, OBC, and GPTQ make no claims that such omissions introduce significant errors; they merely maintain the full Hessian structure during training.
>
> > **W2 & Q2:** The “position-invariant distributions” property is insufficient validation across models, datasets, classes, or conditioning states.
>
> We thank you for your constructive comments.
>
> Following your suggestion, we have conducted **a systematic visualization analysis of ARVG’s token-wise activations**, which has been included in **Appendix O** for reference. We kindly invite you to consult it. The results indicate that ARVG’s position-invariant activation distribution remains consistent across different models, classes, and conditions.
>
> > **W3 & Q3:** Baseline comparisons and fairness of hyperparameter / implementation choices are unclear.
>
> We sincerely appreciate your constructive comments.
>
> The “default settings” mentioned in our paper indicate that we strictly follow the official implementations of the baselines, including both training and quantization configurations.
>
> Following your suggestion, **we have added the complete implementation details for all baselines, along with the configuration table**. Please refer to **Appendix P**.
>
> > **W4:** Some result presentations and tables lack readability and consistency.
>
> Thank you for your detailed comments.
>
> Regarding the collapse of the IS metric for SVDQuant, we provided an explanation in Sec. 5.1.
> The performance degradation of some baselines in Tables 2 and 3 is due to their limited adaptability to ARVG’s unique challenges. For instance, SmoothQuant fails on PAR because its scaling strategy cannot address the extreme and highly dynamic token-wise activations.
>
> For the table headers, sampling batch size and sequence length do not affect model performance and are therefore not explicitly reported. The calibration set and quantized components are introduced in the quantization settings of paper.
>
> > **W5 & Q4:** The method fails to preserve accuracy compared to full precision at 8 bits.
>
> Thank you for your valuable comments.
>
> **ARVG models exhibit higher sensitivity to quantization compared with LLMs and DMs, due to their wider activation ranges and dynamic nature of token dimensions**.
>
> Specifically:
> - Compared with LLMs, whose activations in most layers remain within single-digit magnitudes, the activations of ARVG models often reach several tens or even hundreds.
> - Compared with DMs, the autoregressive nature of ARVG induces highly dynamic token-wise activation across long sequences.
>
> Therefore, achieving nearly lossless INT8 quantization for ARVG is more difficult. In the INT8 setting, our method limits FID degradation to within **0.15** across all MAR models and consistently below **0.25** for all RAR models. Visualizations in **Appendix K** further demonstrate that our method preserves strong semantic consistency and visual fidelity on both PAR and VAR models.
>
> Additionally, we discuss performance degradation under the INT4 setting in the Appendix Limitations. We validate **the effectiveness of our method under the W4A8 setting**, as detailed in **Appendix F, Table 9**.

---

> ### Author Response · Authors · 2025-11-19
>
> > **Q5:** How Formula (16) avoids division-by-zero to ensure the numerical stability of GPS ?
>
> Thank you for your careful comment.
>
> GPS is computed over 128 calibration samples, each containing hundreds of tokens, resulting in **tens of thousands of activation values per channel**. At this scale, the probability of **all activation quantization errors being exactly zero** is statistically negligible.
> We recognize that this constitutes a potential edge case in implementation and sincerely thank you for pointing this out. To address it, we will introduce a safety check: if this rare condition occurs, the scaling factor will be set to 1 to prevent numerical instability.
>
> > **Q6:** How does STWQ handle variable sequence lengths or arbitrary cropping ?
>
> We appreciate your valuable question.
>
> ARVG models have **fixed sequence lengths and input resolutions**, eliminating the need to handle variable sequence lengths and arbitrary cropping.
> STWQ is explicitly designed around this property, enabling fully offline calibration of static token-wise quantization parameters.
>
> For a new ARVG model, STWQ need to rebuild its static parameter table. Because adaptive schemes typically incur additional overhead, no existing method simultaneously achieves adaptivity and fully static deployment.
>
> > **Q7:** How was the threshold of DGC chosen ?
>
> Thank you for your insightful question.
>
> The threshold follows the heuristic used in Q-Diffusion [1], which retains 50% of the timestep samples as the calibration. Our experiments show that this choice remains stable across models and datasets, requiring no additional tuning.
> Although entropy–elbow selection is theoretically more adaptive, it provides limited control over the calibration size: obtaining a fixed size of calibration (e.g., 128 samples) requires repeated entropy estimation and multiple sampling rounds, resulting in substantial overhead.
>
> In response to your question, we conducted an **ablation study on the DGC threshold in Appendix Q**, and the results confirm that the 50% setting yields better performance.
>
> [1] Q-diffusion: Quantizing diffusion models. CVPR 2023.

---

> > ### Comment · Reviewer_oG1L · 2025-11-25
> >
> > Thank you for your response. Although you addressed many points, several of my concerns remain unresolved.
> >
> > ### W1 and Q1
> >
> > 1. First, prior work beginning with OBS clearly states that cross terms cannot be omitted in the same way as OBD. This is discussed on the right half of page 1 in the original paper. Your statement that “OBS, OBC, and GPTQ make no claims that such omissions introduce significant errors; they merely maintain the full Hessian structure during training” is therefore inaccurate.
> > 2. The proof in Appendix A is unrelated to ARVG. Moreover, well-established strategies such as GPTQ and its variants have shown that such omissions are not valid in practice. For this reason, I believe the theoretical justification provided is too loose and cannot reliably guide optimization.
> > 3. Given these points, I would like the authors to explain why this omission does not lead to degradation in ARVG. In addition, I would like to know how much better the results would be if the omission were not applied.
> >
> > ### W4
> >
> > 1. The phrase “likely because low-rank weight decomposition impairs model diversity” requires further explanation. Why does this issue not appear in the non-ARVG setting in the original paper?
> > 2. SVDQuant should not perform worse than its non–low-rank counterpart. Is there any baseline that can support or explain the unusual behavior you observed with SVDQuant?
> >
> > ### W5 and Q4
> >
> > 1. I still believe that the degree of accuracy degradation at W8A8 is difficult to accept.
> > 2. As a fallback, I would like to see the results for weight-only quantization under W4A16 or W8A16. These settings would not suffer from the activation-side dynamic range issues you mentioned.

---

> > > ### Author Response · Authors · 2025-11-27
> > > **Further Response to Reviewer oG1L**
> > >
> > > Dear oG1L:
> > >
> > > Thank you for your proactive response.
> > > We sincerely appreciate the opportunity to further address your concerns and are glad to engage in deeper discussion with you.
> > > We address your concerns as follows:
> > >
> > > ---
> > >
> > > ### W1 and Q1
> > >
> > > > 1: Prior work beginning with OBS clearly states that cross terms cannot be omitted in the same way as OBD.
> > >
> > > OBS states that “For computational simplicity, OBD assumes that the Hessian matrix is diagonal; in fact, however, Hessians for every problem we have considered are strongly non-diagonal, and this leads OBD to eliminate the wrong weights.”
> > > Nevertheless, **we believe that such non-diagonality of Hessian matrix does not compromise the effectiveness of using a diagonal approximation to estimate quantization error.**
> > > Our reasoning is based on the following two considerations:
> > >
> > > - OBD and OBS are pruning methods, where the pruning error is expressed as
> > > $\Delta w = −w$, meaning the entire weight is set to zero.
> > > In contrast, the quantization error $\Delta w = w−w_q$ arises from discretizing the weight and is typically treated as zero-mean noise.
> > > As shown in our proof (Eq. 32(b)), **the cross terms can be neglected not due to any diagonal assumption on the Hessian, but because the quantization error is zero-mean, which leads the expectations of all related cross terms to zero.**
> > > In contrast, pruning errors do not satisfy the zero-mean assumption,
> > > and **quantization errors remain unaffected in expectation by the Hessian’s non-diagonal components.**
> > >
> > > - OBQ and GPTQ mitigate the impact of quantization error by compensating with the remaining weights. Establishing this compensation relies on the non-diagonal structure of the Hessian, **implying that OBQ and GPTQ must operate with the full Hessian matrix**.
> > > However, we believe **this does not imply that using a diagonal Hessian approximation to estimate quantization error, i.e., ignoring cross terms, would “introduce significant errors’’ in quantization-error estimation.**
> > > After reviewing the original OBQ and GPTQ papers, we also did not find statement suggesting that “such omissions introduce significant errors.’’
> > >
> > > In summary, we view these as two distinct ways of leveraging the Hessian.
> > > OBQ, GPTQ, and related methods rely on its non-diagonal structure to model the compensation between quantized and remaining weights, adjusting the latter ***to mitigate quantization error.***
> > > In contrast, our method requires only a diagonal Hessian approximation ***to estimate the quantization error itself.*** **As the goals and required information differ, the two are not in conflict.**
> > >
> > > > 2 and 3:
> > > GPTQ and its variants have shown that such omissions are not valid in practice. How much better the results would be if the omission were not applied.
> > >
> > > Appendix A provides a general theoretical analysis of activation–weight quantization.
> > > In particular, **Appendix A.3 shows that, due to the zero-mean property of rounding-to-nearest quantization error, the cross terms can be safely ignored in expectation.**
> > > **Appendix L** further confirms that cross terms are numerically negligible in practice, empirically justifying their omission.
> > >
> > > To further address your concern, we re-derive GPS as follows:
> > >
> > > Retaining the cross terms, Eqs. 11–12 can be rewritten as:
> > >
> > > $
> > > E_{\mathbf{x}}' \approx \frac{1}{2}\mathbb{E}\left[W_{1,1}^2 \Delta x_1^2 + \frac{2s_2}{s_1} W_{1,1} W_{2,1} \Delta x_1 \Delta x_2  + \frac{s_2^2}{s_1^2} W_{2,1}^2 \Delta x_2^2 \right]
> > > $
> > >
> > > $
> > > E_{\mathbf{W_{:,1}}}'
> > >     \approx \frac{1}{2} \mathbb{E}\left[ \Delta {W_{1,1}}^2{x_1}^2 + \frac{2s_1}{s_2}\Delta {W_{1,1}} \Delta {W_{2,1}} x_1 x_2 + \frac{{s_1}^2}{{s_2}^2}\Delta {W_{2,1}}^2{x_2}^2 \right]
> > > $
> > >
> > > The scaling gain function in Eq. 13 becomes:
> > >
> > > $
> > > g(s_2) = \frac{1}{2} ( {W_{2,1}}^2\Delta {x_2}^2 + \Delta {W_{2,1}}^2{x_2}^2 - \frac{{s_2}^2}{{s_1}^2}{W_{2,1}}^2\Delta {x_2}^2 - \frac{{s_1}^2}{{s_2}^2}\Delta {W_{2,1}}^2{x_2}^2 ) +  (\frac{{s_1}-s_2}{s_1} W_{1,1} W_{2,1} \Delta x_1 \Delta x_2 - \frac{{s_1}-s_2}{s_2}\Delta {W_{1,1}} \Delta {W_{2,1}} x_1 x_2)
> > > $
> > >
> > > Eq. 14 can be rewritten as:
> > >
> > > $
> > > g'(s_2) = - \frac{{s_2}}{{s_1}^2}{W_{2,1}}^2\Delta {x_2}^2 + \frac{{s_1}^2}{{s_2}^3}\Delta {W_{2,1}}^2{x_2}^2 +(\frac{-1}{s_1} W_{1,1} W_{2,1} \Delta x_1 \Delta x_2 + \frac{{s_1}}{{s_2}^2}\Delta {W_{1,1}} \Delta {W_{2,1}} x_1 x_2)
> > > $
> > >
> > > As shown, $s_2$ admits four solutions, and the same procedure can be extended to Eq. 15. We compute all valid solutions and choose the one that maximizes the gain in Eq. 13 as the GPS scaling factor when cross terms are included, which introduces additional overhead compared to original GPS.
> > > **We represent it as GPS-full and evaluates its difference from the original method in terms of quantization error, as shown in Appendix N.** The results show that retaining cross terms has a negligible impact on original GPS.

---

> > > ### Author Response · Authors · 2025-11-27
> > > **Further Response to Reviewer oG1L**
> > >
> > > ### W4
> > >
> > > > 1: Why does this issue not appear in the non-ARVG setting in the original paper?
> > >
> > > Thank you for pointing out the anomaly in the IS metric of SVDQuant. Our investigation confirms that the issue originates from **a known bug in the evaluation toolkit** (see the [official issue](https://github.com/openai/guided-diffusion/issues/153
> > > )). When images are generated and fed into the evaluator in ImageNet class order, the IS score is incorrectly fixed at approximately 50.
> > > We re-evaluated and corrected the SVDQuant IS results for all VAR models. We appreciate your careful and helpful comments.
> > >
> > > > 2: SVDQuant should not perform worse than its non–low-rank counterpart.
> > >
> > > SVDQuant performs strongly on diffusion models primarily because its low-rank decomposition effectively mitigates activation outliers in the qkv and fc1 layers. However, the method **keeps the quantization-sensitive proj layers and KV projections in full precision**.
> > > In our setting, we employ quantization for all layers and QKV projections in ARVG. To ensure fairness, we also apply SVDQuant to all layers and QKV projections. As discussed in the paper, **ARVG exhibits stronger token-wise activation dynamics in the proj and fc2 layers**, which leads to the degraded performance of SVDQuant on ARVG.
> > >
> > > Moreover, SVDQuant is tailored to the activation characteristics of diffusion models, which limits its applicability to autoregressive architectures. Existing quantization studies (e.g., FBQuant[1]) also report that **SVDQuant underperforms non–low-rank methods such as OmniQuant and AWQ across LLM's evaluations**, further indicating its limited cross-architecture adaptability.
> > >
> > > Therefore, due to the distinct activation distributions and autoregressive architecture of ARVG, SVDQuant cannot retain the advantages it demonstrates on diffusion models.
> > >
> > >
> > > [1] Fbquant: Feedback quantization for large language models.
> > >
> > >
> > > ### W5 and Q4
> > >
> > > > The degree of accuracy degradation at W8A8 is difficult to accept. I would like to see the results for weight-only quantization under W4A16 or W8A16.
> > >
> > > Following your suggestion, we added weight-only experiments to further validate the challenge of activation quantization in ARVG.
> > >
> > > We employ SmoothQuant to several models under the W8A16 and W6A16 settings (weight-only quantization).
> > > As shown in Table, the accuracy is almost unaffected.
> > > However, **when activation quantization is applied, W8A8 still causes severe performance degradation across all models, even with SmoothQuant’s outlier smoothing**.
> > > In contrast, SmoothQuant shows nearly no degradation in W8A8 quantization for diffusion models [2] and LLMs [3].
> > > Thus, compared with prior models, ARVG exhibits stronger activation outliers and greater token-wise dynamics, making activation quantization substantially more challenging.
> > >
> > > To address these ARVG-specific challenges, we introduce the first activation–weight quantization method tailored for ARVG. **Experimental results demonstrate that our method substantially outperforms existing approaches and is nearly lossless: FID drops are <0.1 for MAR, <0.25 for RAR, and <1.0 for most PAR and VAR models.**
> > > We hope our work provides insights for future ARVG quantization research, and we will continue to pursue deeper exploration in this direction.
> > >
> > > |Model|Bits|IS|FID|sFID|
> > > |----|----|----|---|---|
> > > |VAR-d16|W16A16|283.21|3.60|8.27|
> > > | |W8A16|282.23|3.46|8.33|
> > > | |W8A8|229.87|4.29|13.39|
> > > | |W6A16|280.37|3.39|8.49|
> > > | |W6A6|101.55|18.54|17.22|
> > > |RAR-B|W16A16|292.80|1.96|6.16|
> > > | |W8A16|289.45|2.00|6.27|
> > > | |W8A8|242.97|2.80|7.76|
> > > | |W6A16|267.71|2.63|7.11|
> > > | |W6A6|31.04|63.77|42.08|
> > > |PAR-XL|W16A16|259.20|2.61|6.30|
> > > | |W8A16|256.88|2.58|6.17|
> > > | |W8A8|8.28|132.17|79.50|
> > > | |W6A16|236.36|2.59|5.80|
> > > | |W6A6|8.08|146.52|69.36|
> > >
> > > [2] Vidit-q: Efficient and accurate quantization of diffusion transformers for image and video generation.
> > > [3] Smoothquant: Accurate and efficient post-training quantization for large language models.

---

> > > > ### Comment · Reviewer_oG1L · 2025-11-28
> > > >
> > > > Thank you again for your responses. I now fully understand both the SlimDiff methodology and the issues concerning the baseline comparisons.
> > > >
> > > > However, I still believe W8A8 causes noticeable degradation that undermines the practicality of quantization on ARVG models. I will therefore **raise my rating to 6**, but no higher. It appears the edit button to change my rating is no longer available, so I hope the Area Chair will take my updated comments into account.

---

> > > > > ### Author Response · Authors · 2025-12-03
> > > > > **Response to Reviewer oG1L**
> > > > >
> > > > > Dear oG1L:
> > > > >
> > > > > Thank you for your positive response.
> > > > >
> > > > > We understand that perspectives on what constitutes noticeable degradation may vary.
> > > > > Nevertheless, the empirical results indicate that our method remains highly practical under 8-bit quantization: the visualizations in Appendix K show that the quantized and full-precision models produce images with closely matched semantics and perceptual quality; moreover, prior quantization studies [1,2,3] on generative models consistently report that such minor FID drops fall well within the commonly accepted range.
> > > > >
> > > > > ###### [1] TFMQ-DM: Temporal Feature Maintenance Quantization for Diffusion Models, CVPR2024 Highlight
> > > > > ###### [2] Quest: Low-bit diffusion model quantization via efficient selective finetuning, ICCV 2025
> > > > > ###### [3] Q-dit: Accurate post-training quantization for diffusion transformers, CVPR2025

---

### Official Review · Reviewer_D8Am · 2025-11-09

**Soundness:** 2
**Presentation:** 2
**Contribution:** 2
**Rating:** 4
**Confidence:** 4

**Summary:**

This paper proposes several techniques to improve the quantization of autoregressive visual generation (ARVG) models. The authors identify that a key challenge in quantizing ARVG lies in its highly skewed and dynamic activation distributions. To address this, they introduce a Gain-Projected Scaling (GPS) mechanism that stabilizes quantization and better preserves generative performance. Experimental results demonstrate substantial improvements over existing quantization baselines across multiple ARVG architectures.

**Strengths:**

* The proposed Gain-Projected Scaling (GPS) offers a principled mechanism for balancing weight and activation scaling by leveraging Hessian information rather than relying on empirical heuristics. This theoretically grounded design suggests strong potential for broader applicability beyond ARVG models.

* Experimental results consistently show that the proposed method achieves significant improvements over baseline quantization approaches, underscoring its effectiveness and practical value.

**Weaknesses:**

* Achieving the optimal GPS configuration appears challenging compared to empirical estimation. The approach involves several approximations, including those for the Hessian matrix, the upper bound of overall quantization loss, and scaling quantization error. These approximations may introduce discrepancies across different input distributions and norms. Moreover, the final GPS algorithm computes scaling based only on the most significant channel, which may not reflect a global—or even local—optimum. It would be helpful if the authors could include ablation studies illustrating how quantization error changes with GPS.

* The methodological presentation lacks clarity, particularly in Section 4.3, which is relatively brief and insufficiently detailed. The notion of mismatched calibration requires a clearer explanation, as does the proposed distribution-guided calibration and how it specifically resolves this issue.

* The paper does not clearly articulate the unique challenges of quantizing ARVG models. Dynamic ranges and channel-wise outliers are common issues in diffusion model quantization, and the proposed GPS appears to be a general technique rather than one specifically tailored for ARVG. Similarly, the STWQ module seems more like a heuristic quantizer selection strategy based on data distribution. The authors should clarify what makes ARVG quantization distinct and how their method specifically addresses those challenges.

* Empirically, the method shows only marginal improvements except on VAR and PAR models. It would be helpful to clarify whether this is related to baseline implementation or model-specific characteristics.

**Questions:**

Please refer to the weakness.

---

> ### Author Response · Authors · 2025-11-19
>
> Many thanks for the reviewer's valuable comments, which help us a lot to improve our work. We address your concerns as follows.
>
> ---
>
> > **W1.1:** Several approximations in GPS may introduce discrepancies across different input distributions and norms.
>
> Thank you for your professional comment.
>
> GPS relies on a assumption (Remark 1). As detailed in Sec. 4.1 of the paper, we have **validated this assumption via comprehensive statistical analyses across multiple ARVG models**.
>
> In addition, GPS involves three approximations:
>
> - Hessian approximation of activation–weight quantization loss. We approximate it using an MSE loss and neglect the cross terms.
> The theoretical justification for this approximation is provided in **Appendix A.3**.
>
> - Upper-bound approximation of overall quantization error. We approximate the activation quantization error on the weight-quantized model using the activation quantization error on the full-precision model.
> Its theoretical justification is provided in **Appendix A.4**.
>
> - Scaling quantization error approximation. We estimate the activation quantization error after scaling based on inter-channel scaling effects, which serves as an approximation of the true quantization error after scaling.
> This approximation **builds upon the error–estimation strategy validated in DilateQuant**.
>
> Following your suggestion, we quantify the biases of these three approximations across different ARVG models, layers, and input distributions.
> The results show that **all biases remain consistently negligible across all settings**.
>
> The detailed results have been added to **Appendix L**, and we kindly invite you to review them.
>
> > **W1.2:** GPS may not reflect a global optimum. It would be helpful if the authors could include ablation studies illustrating how quantization error changes with GPS.
>
> Thank you for your valuable comment.
>
> In GPS, the quantization-sensitive maximal channel is used as a reference to derive optimal scaling factors for all other channels.
> Ablation studies in the paper confirm that GPS outperforms existing scaling methods (Table 6), and **in randomized experiments, it achieves the best overall performance (Figure 5)**.
>
> Following your suggestion, we further validate the advantages of GPS from two perspectives:
> - **Ablation study on channels used to compute the scaling factor.**
>    This demonstrates the superiority of selecting the maximal channel (please see **Appendix M**).
> - **Visualization of GPS’s impact on quantization error.**
>    This confirms its effectiveness in reducing the error (please see **Appendix N**).
>
> > **W2:** The methodological presentation lacks clarity in Section 4.3.
>
> Thank you for your constructive suggestion.
>
> **Mismatch calibration** refers to the sample redundancy in ARVG, **which causes the quantization process to overemphasize high-density activations while underrepresenting diverse activations**. This results in quantization parameters that are misaligned with the true distribution. This phenomenon is illustrated in our activation distribution visualizations (Fig. 1(b) of the paper).
> More specifically, the activations of **unconditional samples** exhibit highly similar distributions and dominate the overall sample distribution, causing the quantization range to be largely determined by these redundant activations, thereby reducing the precision allocated to **conditional activations**.
>
> DGC addresses this issue by leveraging the Mahalanobis distance to **estimate distributional entropy, which jointly captures both sample diversity and density**. For each sample batch, DGC selects the **top 50% of samples with the highest entropy** to construct the calibration.
> Specifically, to obtain a calibration set of 128 samples, we set the model’s batch size to 128, yielding 256 samples in total (128 conditional and 128 unconditional). We then compute the distributional entropy of all 256 samples using DGC and select the top 50% to form the calibration.
>
> Due to space constraints, Section 4.3 was necessarily condensed, which may have introduced some ambiguity. We will present a more comprehensive and clearer version in the final manuscript.

---

> ### Author Response · Authors · 2025-11-19
>
> > **W3:** The authors should clarify what makes ARVG quantization distinct and how their method addresses those challenges.
>
> Thank you for your constructive comment.
>
> **Quantization distinct of ARVG:** Compared with LLMs, ARVGs exhibit larger activation ranges and more frequent outliers due to their visual tokens. Compared with DMs, the autoregressive paradigm induces highly dynamic, token-wise activation distributions.
>
> - To address the wider activations, unlike the empirically designed scaling method in DMs, GPS is the first to **theoretically quantify the effect of scaling on quantization error and to derive an optimal scaling strategy**.
>
> - To address token-wise dynamic activations, unlike dynamic quantization based on min–max calibration, STWQ leverages the fixed token lengths and position-invariant distributions of ARVG to introduce a **high-precision, static token-wise calibration scheme**.
>
> Ablation studies and statistical analyses in the paper demonstrate that GPS and STWQ effectively address these unique challenges of ARVG.
>
> > **W4:** The method shows marginal improvements except on VAR and PAR models.
>
> Thank you for your valuable suggestion.
>
> Our improvements on VAR and PAR are smaller than those achieved by RAR and MAR. We believe the reviewer may have inadvertently mistyped the word “expect” in their comment.
>
> Under the VAR-16 and VAR-d24 INT6 settings, our method surpasses the strongest baseline, QuaRot, by 0.62 and 0.39 FID, respectively. However, as emphasized in Appendix G, QuaRot provides **no practical speedup** for VAR since its rotation factors cannot be absorbed offline. Consequently, our method achieves **actual improvements of 4.19 and 0.72 FID** over the strongest practical baseline, SVDQuant.
> This represents a significant improvement compared with most existing quantization methods [1,2,3] for image generation.
> Under the PAR-XL and PAR-XXL INT6 settings, our method achieves improvements of **6.65 and 7.75 FID** over the strongest baseline, SVDQuant.
>
> In addition, the baseline methods were implemented strictly following their open-source code, with comprehensive implementation details included in **Appendix P**.
>
> [1] TFMQ-DM: Temporal Feature Maintenance Quantization for Diffusion Models, CVPR2024 Highlight
> [2] Quest: Low-bit diffusion model quantization via efficient selective finetuning, ICCV 2025
> [3] Q-dit: Accurate post-training quantization for diffusion transformers, CVPR2025

---

### Author Response · Authors · 2025-12-03
**Rebuttal Summary of Submission 11681 - Page 2 / 2**

> **concern 1**: 8-bit quantization causes noticeable degradation

Based on prior experience, the reviewers (oG1L, FANZ) expected ARVG models to remain nearly lossless under 8-bit activation–weight quantization.
To address this concern, we clarify the following points:

- **ARVG models face more severe quantization challenges than prior models.** For instance, under 8-bit settings, SmoothQuant achieves near-lossless performance on large language models and diffusion models. However, on ARVG, its performance degrades significantly: FID for RAR-XL model drops by 1.81, and PAR models even experience performance collapse.
- **We have evaluated state-of-the-art quantization methods on ARVG**, including the training-based OmniQuant, the low-rank decomposition SVDQuant, and the rotation-based QuaRot. The results show that even these advanced methods suffer significant performance degradation under 8-bit settings, further highlighting the difficulty of ARVG quantization.
- **Our method consistently outperforms the state-of-the-art baselines across diverse ARVG models**, demonstrating its effectiveness and specificity for ARVG. This has also been acknowledged by all reviewers.
- **Our method achieves nearly lossless performance under the 8-bit setting: FID drops are <0.1 for MAR, <0.25 for RAR, and <1.0 for most PAR and VAR models.**
Evidence from existing quantization works [1,2,3] confirms that such drops are minimal and fully acceptable.


##### [1] TFMQ-DM: Temporal Feature Maintenance Quantization for Diffusion Models, CVPR2024 Highlight
##### [2] Quest: Low-bit diffusion model quantization via efficient selective finetuning, ICCV 2025
##### [3] Q-dit: Accurate post-training quantization for diffusion transformers, CVPR2025

> **concern 2**: Several approximations in GPS are not sufficiently validated.

- Reviewer D8Am expressed concern that “these approximations may introduce discrepancies across different input distributions and norms.”
- Reviewer oG1L expressed concern that “GPS omits Hessian cross terms in its derivation. prior works pointed out that such omissions can introduce significant errors.”

To address these concerns, on the one hand, we clarify that **all approximations in our method are grounded in rigorous theory**:
- The proof for Hessian approximation of quantization loss is provided in Appendix A.3
- The proof for upper-bound approximation of quantization error is provided in Appendix A.4
- The proof for scaling quantization error approximation builds upon the error–estimation strategy validated in DilateQuant.

On the other hand, we provide **additional extensive empirical evidence in the rebuttal**.
- In response to D8Am’s concern, we conduct a **systematic statistical evaluation of the three approximation errors in Appendix L**, covering diverse data distributions, models, and network layers. The results show that these errors remain consistently minimal across all conditions, confirming the stability and robustness of the approximations.
- In response to oG1L’s concern, we clarify that prior works retained the Hessian cross-terms because they were essential for modeling inter-weight compensation, and they did not point out that such omissions can introduce significant errors. In contrast, our method can safely ignore these terms due to the zero-mean property of quantization errors. **These two lines of research are therefore fundamentally unrelated.**
We also provide **statistical evaluations of these cross-terms in Appendix L and assess their impact on GPS in Appendix N**. The results show that omitting the cross-terms is fully justified.


Although Reviewer D8Am did not respond, we believe that the experiments and explanations in our rebuttal can address his concerns. Through two rounds of discussion with Reviewer oG1L, we were encouraged by his comment, ***“I now fully understand the methodology. I will raise my rating to 6.”***
Unfortunately, while our discussion with Reviewer FANZ was highly constructive, the compressed rebuttal schedule precluded his final feedback. Considering the clarity and convergence already achieved, we believe that further discussion would likely have led to an agreement if the discussion period had not been cut short.

---

Finally, we again thank all reviewers for their comments and especially thank the ACs for their effort and time.

---

### Author Response · Authors · 2025-12-03
**Rebuttal Summary of Submission 11681 - Page 1 / 2**

We sincerely appreciate the reviewers’ professional and valuable comments. Below is a summary of the discussions during the rebuttal phase.

We note that the reviewers **consistently recognized the novelty of our work**, as reflected in comments such as *“theoretically grounded design”* (D8Am), *“more convincing than many scaling methods”* (oG1L), *“solid theoretical and methodological innovation”* (FANZ), and *“provides a theory-driven scaling solution rather than heuristics”* (NhoA).
Meanwhile, **the thorough experiments and effectiveness validation of our method were recognized** by the reviewers, such as *“underscoring its effectiveness and practical value”* (D8Am), *“comprehensive experimental coverage and ablation”* (oG1L), *“consistent superiority over existing PTQ methods”* (x7xs), and *“comprehensive experimental validation”* (FANZ).
Moreover, the reviewers **acknowledged the clear motivation and comprehensive analysis of ARVG unique quantization challenges in our work**, such as *“Problem formulation is clear and well targeted”* (oG1L), *“Clarity and Technical Detail”* (x7xs), *“deeply analyzes and identifies three specific, critical challenges”* (FANZ), and *“Clear motivation with three unique quantization challenges”* (NhoA).
Furthermore, the reviewers **acknowledged the significance and value of our contributions**. In particular, reviewer FANZ commented that the *“This is a solid and elegant theoretical contribution.”*

Additionally, the reviewers also offered several constructive suggestions.
Following these suggestions, we explained the baseline reproduction details and further introduced the depth analyses and implementation procedures in the original Appendix C and D for Sections 4.2/4.3.
**These responses have been agreed upon by reviewers oG1L, FANZ, and NhoA.**
Furthermore, the reviewers' concerns generally fall into **two issues**, to which we provided detailed responses via supplementary experiments and clarifications.
Details are as follows:

---

### Meta-Review · Area_Chair_6KWt · 2026-01-18

**Summary:**

The paper received a mix of borderline reviews. The main initial concerns were (i) the degree of degradation observed, (ii) the performance improvement is a bit marginal, and (iii) the paper clarity. The rebuttal addressed most of the reviewers' concerns, including some of the major ones. At least one initially negative reviewer explicitly said they would raise the score to 6. Considering the clear motivation and novelty of the proposed method, as well as the reasonable empirical evidence, I recommend acceptance of the paper.

**Reviewer Concerns:**

Most concerns were addressed by the rebuttal. The one remaining question is whether the observed degradation is acceptable; this is inherently a case-by-case trade-off between quality and computation budgets, rather than a methodological gap. Overall, there are no clear outstanding issues that would prevent acceptance.

**Reviewer Scores:**

The scores are mixed and largely borderline. One reviewer explicitly stated they would raise their score to above the acceptance threshold after the rebuttal.

---

### Decision · Program_Chairs · 2026-01-26

Accept (Poster)